# Simulating organic aerosol in Delhi with WRF-Chem using the VBS approach: Exploring model uncertainty with a Gaussian Process emulator

Ernesto Reyes-Villegas[1,a], Douglas Lowe[1,b], Jill S. Johnson[2,c], Kenneth S. Carslaw[2], Eoghan Darbyshire[1,d], Michael Flynn[1], James D. Allan[1,e], Hugh Coe[1], Ying Chen[3], Oliver Wild[3], Scott Archer-Nicholls[4], Alex Archibald[4], Siddhartha Singh[5], Manish Shrivastava[6], Rahul A. Zaveri[6], Vikas Singh[7], Gufran Beig[8], Ranjeet Sokhi[9], Gordon McFiggans[1]

[1]Department of Earth and Environmental Sciences, University of Manchester, Manchester, M13 9PL, UK
[2]Institute for Climate and Atmospheric Science, School of Earth and Environment, University of Leeds, Leeds, LS2 9JT, UK
[3]Lancaster Environment Centre, Lancaster University, Lancaster, LA1 4YQ, UK
[4]NCAS-Climate, Department of Chemistry, University of Cambridge, Cambridge, CB2 1EW, UK
[5]Ozone Unit, India Meteorology Department, New Delhi, 110003, India
[6]Atmospheric Sciences & Global Change Division, Pacific Northwest National Laboratory, Richland, Washington 99352, United States
[7]National Atmospheric Research Laboratory, Gadanki, AP, India
[8]Indian Institute of Tropical Meteorology, Pune, 411008, India
[9]Centre for Atmospheric and Climate Physics Research, University of Hertfordshire, Hertfordshire, AL10 9AB, UK
[a] now at: Tecnologico de Monterrey, Escuela de Ingeniería y Ciencias, Av. General Ramon Corona 2514, Nuevo Mexico, Zapopan CP 45201, Jalisco, Mexico.
[b] now at: IT Services, University of Manchester, Manchester, M13 9PL, UK
[c] now at University of Sheffield, Sheffield, S10 2TN, UK
[d] now at: The Conflict and Environment Observatory, Hebden Bridge, HX7 5HZ, UK
[e] National Centre for Atmospheric Science, The University of Manchester, Manchester, M13 9PL, UK

*Correspondence to*: Gordon McFiggans (g.mcfiggans@manchester.ac.uk)

**Abstract.** The nature and origin of organic aerosol in the atmosphere remain unclear. The gas-particle partitioning of semi-volatile organic compounds (SVOC) that constitute primary organic aerosols (POA) and the multigenerational chemical aging of SVOCs are particularly poorly understood. The volatility basis set (VBS) approach, implemented in air quality models such as WRF-Chem, can be a useful tool to describe emissions of POA and its chemical evolution. However, the evaluation of model uncertainty and the optimal model parameterisation maybe expensive to probe using only WRF-Chem simulations. Gaussian process emulators, trained on simulations from relatively few WRF-Chem simulations, are capable of reproducing model results and estimating the sources of model uncertainty within a defined range of model parameters. In this study, a WRF-Chem VBS parameterisation is proposed; we then generate a perturbed parameter ensemble of 111 model runs, perturbing ten parameters of the WRF-Chem model relating to organic aerosol emissions and the VBS oxidation reactions. This allowed us to cover the model's uncertainty space and compare output from each run to aerosol mass spectrometer observations of organic aerosol concentrations and O:C ratios measured in New Delhi, India. The simulations spanned the organic aerosol concentrations measured with the AMS. However, they also highlighted potential structural errors in the model that may be related to unsuitable diurnal cycles in the emissions and/or failure to adequately represent the dynamics of the planetary boundary layer. While the structural errors prevented us from clearly identifying an optimised VBS approach in WRF-Chem, we were able to apply the emulator in two periods: the full period (1st -29th May) and a subperiod period 14:00-16:00 hrs local time, 1st-29th May. The combination of emulator analysis and model evaluation metrics allowed us to identify

plausible parameter combinations for the analysed periods. We demonstrate that the methodology presented in this study can
be used to determine the model uncertainty and identify the appropriate parameter combination for the VBS approach, and
hence provide valuable information to improve our understanding on OA production.

## 1 Introduction

Over the last decades, India has been facing air pollution problems and is ranked fifth in the 2020 world air quality ranking
(IQair, 2021) and Delhi ranked as one of the most polluted cities in the world with related health burden of about 10,000
premature deaths annually (Chen et al., 2020a), based on $PM_{2.5}$ measurements (particulate matter lower than 2.5 micrometers
in diameter). This situation has a remarkable impact on Indian citizens due to India having a population that is larger than one
billion inhabitants.
Organic aerosols (OA) are one of the main constituents of submicron particulate matter, accounting for between 20% – 90%
of the total aerosol mass concentration in urban environments (Kanakidou et al., 2005;Zhang et al., 2007).Various studies have
been performed in India looking at the particulate matter composition and source identification of OA using receptor modelling
tools (Kompalli et al., 2020;Jain et al., 2020;Cash et al., 2021;Reyes-Villegas et al., 2021) along with investigating the health
risks associated with aerosols (Shivani et al., 2019;Gadi et al., 2019). However, one limitation of receptor models is that they
do not involve chemical processing. The use of regional atmospheric models allows the study of the temporal and spatial
behaviour of various chemical species of OA. The Weather Research and Forecasting model coupled with Chemistry (WRF-
Chem) is a regional 3-D atmospheric model that simulates the emissions and dispersion of gaseous and particulate species,
including the chemical processes and their interaction with meteorology. There have been recent WRF-Chem studies
investigating $PM_{2.5}$ concentrations (Bran and Srivastava, 2017;Chen et al., 2020b;Jat et al., 2021; Ghosh et al., 2021) and
volatile organic compounds (VOC) (Chutia et al., 2019) over India.
Despite the recent studies on aerosol sources and processes involving both observations and modelling, there is still a gap
between observations and modelling studies, for example with particulate organic matter being generally underestimated by
models (Bergström et al., 2012;Tsigaridis et al., 2014), mainly attributed to the lack of understanding of the emission sources,
the OA processes and SOA mechanisms. Hence, we need to understand the capability of organic matter to produce and retain
fine particulate mass in order to fully understand their processes and impacts on air quality and climate (Carlton et al., 2010;von
Schneidemesser et al., 2015). It is here where the volatility basis set (VBS) scheme can be valuable when implemented in
chemical transport models. The VBS scheme describes the chemical ageing of particulate organic matter, its chemical
processing and associated volatility (Donahue et al., 2006;Shrivastava et al., 2011; Bianchi et al., 2019). It treats POA
emissions as semi volatile and distributes particulate organic matter by its volatility. This distribution, based on their saturation
concentration ($C^*$), includes low volatility (LVOC), semivolatile (SVOC) and intermediate volatility (IVOC) organic
compounds (Tsimpidi et al., 2016). POA constitutes emissions from anthropogenic combustion processes and open biomass
burning (Stewart et al., 2021a;Stewart et al., 2021b) and by being considered to be semivolatile, the initial particulate organic
matter partially evaporates due to atmospheric dilution followed by the oxidation of evaporated semi-volatile organic vapors.
The resulting low volatility oxidized organic vapors can condense to produce secondary organic aerosol (SOA) (Shrivastava
et al., 2008). This favours the formation of IVOCs and SVOCs in the gas phase. Previous studies have found that IVOCs and
SVOCs can act as a reservoir of organic species that are able to repartition to the particle phase after suffering chemical
processing (Robinson et al., 2007;Lane et al., 2008).
Regional (Li et al., 2016;Akherati et al., 2019) and global models (Tsigaridis et al., 2014;Tilmes et al., 2019) have been
successfully used to simulate aerosol dispersion and chemical processing to some extent. However, they can be highly
uncertain (Bellouin et al., 2016;Johnson et al., 2020), particularly when comparing with on-site observations in a high time
resolution. This uncertainty can be due to a wide range of parameter settings, emission sources or missing processes, and is
challenging to comprehensively evaluate by only running direct model simulations, due to the computing time and expense
needed. Statistical analysis to evaluate model performance over parameter uncertainty can be made tractable through the use
of a statistical emulator (Carslaw et al., 2018). With a trained emulator, it is possible to study thousands or millions of model
variants (parameter combinations) and estimate the sources of uncertainty (Lee et al., 2011;Johnson et al., 2018;Wild et al.,
87 2020)

The VBS approach is often tuned to the environment of interest (Bergström et al., 2012;Shrivastava et al., 2013;Tilmes et al.,
2019;Shrivastava et al., 2019;Shrivastava et al., 2022) and, as mentioned before, doing this only with WRF-Chem runs is
particularly challenging and time consuming. The aim of this study is to determine an effective way of tuning the VBS scheme
using observations, and also to learn about the processes controlling OA in Delhi. Hence, we need to explore the combination
of different techniques, i.e., observations, WRF-Chem modelling with VBS implementation and statistical emulators, to better
understand the partitioning of organic matter and the evolution of POA. In this study, a WRF-Chem parameterisation is
proposed to simulate organic mass concentrations and organic to carbon (O:C) ratios over the region of New Delhi, India, that
includes primary and ageing parameters in the VBS scheme. In this parameterisation we explore the perturbation to the chosen
anthropogenic POA and biomass burning POA parameters that would be needed to give the best fit to the observed OA. We
are not perturbing the SOA parameters from the base case nor the dry and wet deposition simulation uncertainties, analysis
that is out of scope of this work. We also appreciate that there will be sensitivity to the deposition rate of OA components. We
have focused our study on the sensitivity of the OA production processes at a constant deposition rate within WRF-Chem
allowing reasonable conclusions about the plausible range of the other parameters to be drawn notwithstanding this limitation.
The model performance is evaluated over a multi-dimensional parameter uncertainty space that explores parameter uncertainty
in these schemes. We generate a perturbed parameter ensemble (PPE) of 111 model runs that cover the model's uncertainty
space and compare output from each run to AMS observations of OA concentrations and O:C ratios measured at New Delhi,
India. The PPE is then used to construct statistical emulators and sample densely over the uncertainty for a more detailed
comparison over a specific time-period of the observations. The evaluation over specific time-periods will allow to study the
behaviour of the model setup under different conditions, i.e., high vs low mass concentrations, and analyse the impact the
different parameter setups have on the organic mass concentrations.

## 108 2 Methodology

### 109 2.1 WRF-Chem parameterisation and setup

The Weather Research and Forecasting model coupled with Chemistry (WRF-Chem) is used to simulate the emission,
transport, mixing, and chemical transformation of trace gases and aerosols concurrently with meteorology data (Grell et al.,
2005; Fast et al., 2006). Here, WRF-Chem version 3.8.1 is run with a 15 km domain, 127 x 127 grid cells, (Figure 1) and a
simulation period from 19[th] April - 29[th] May 2018, with substantial modification, details in below. This period was selected in
order to compare with aerosol measurements performed at New Delhi (Reyes-Villegas et al., 2021). Table 1 lists the
components that contribute to our model set-up, including the chemistry and aerosol schemes, emissions inventories and
boundary condition specifications. Gas-phase chemistry is simulated with the Common Representative Intermediates (CRI)
mechanism which permits a reasonably detailed representation of volatile organic compound oxidation. The aerosol chemistry
is simulated using the sectional MOSAIC module (Zaveri et al., 2008), including $N_2O_5$ heterogeneous chemical reactions
(Archer-Nicholls et al., 2014;Bertram and Thornton, 2009) and is coupled to the aqueous phase, which allows aerosols to act
as cloud condensation nuclei, as well as the removal of aerosols through wet deposition processes. The aerosol size distribution
in MOSAIC is described by eight size bins spanning a dry particle diameter range of 39nm to 10μm (Zaveri et al., 2008).
Table 1: WRF-Chem setup

| Parameter | Set up |
|---|---|
| Gas phase mechanism | CRI-v2R5 (Watson et al., 2008;Archer-Nicholls et al., 2014) |

| | |
|---|---|
| Aerosol module | MOSAIC (Zaveri et al., 2008;Fast et al., 2006) with VBS (Shrivastava et al., 2011) with SOA (Tsimpidi et al., 2010) |
| Anthropogenic emissions | EDGAR-HTAP and SAFAR-India (CRI-v2R5 speciation) |
| Fire emissions | FINN 1.5 (Wiedinmyer et al., 2011) |
| Biogenic emissions | MEGAN 2.04 (Guenther et al., 2006) |
| Chemical Boundaries | CESM2/WACCM (Danabasoglu et al., 2020) |
| Meteorological Boundaries | ECMWF Reanalysis (Hersbach et al., 2018) |


Our main modifications are focused on the treatment of the organic aerosol (OA) components. Primary organic aerosol (POA)
is treated as semi-volatile, using the Volatility Basis Set (VBS) treatment of Shrivastava et al. (2011). Their 9 volatility bin
VBS scheme has been adapted for use in the 8 size bin version of MOSAIC. Secondary organic aerosol (SOA) has been
included based on the scheme described in Tsimpidi et al. (2010), providing 'anthropogenic' (ARO1 and ARO2 in the original
scheme, SAPRC99) and 'biogenic' (isoprene and monoterpenes) SOA components, each covering 4 volatility bins with $C^*$
values (at 298 K) of 1, 10, 100, and 1000 μg.m-3. ARO1 represents the aromatics with OH reaction rates less than $2x10^4$ ppm$^{-1}$
min$^{-1}$, and ARO2 the aromatics with OH reaction rates greater than $2x10^4$ ppm$^{-1}$ min$^{-1}$. In mapping these to the CRI-v2R5
scheme we have used TOLUENE and BENZENE as the precursors for the ARO1 reactions, OXYL (xylene and other
aromatics) for the ARO2 reactions, and APINENE for the monoterpenes. Indicative SOA yields are given in Table S1 in the
supplementary material. Co-condensation of water has been added for these semi-volatile organics, and they have been coupled
to the aqueous phase in the same manner as other aerosol compounds in MOSAIC. Previous studies have demonstrated that
the condensation of semivolatile organic material onto aerosol particles substantially increases the soluble mass of particles,
their chemical composition and eventually their effective dry size (Topping et al., 2013;Crooks et al., 2018). The mapping of
CESM2/WACCM compounds to CRI-v2R5 and MOSAIC components, for the chemical boundaries, is detailed in Table S2
in the supplementary material. A spin-up period of 11 days was used, from 19th April to 1st May. The meteorological driving
fields were taken from ERA5 reanalysis data. Spectral nudging of the uv wind parameters, temperature and geopotential height
variables to these, above model level 18 and for wavelengths greater than 950km, was used. The domain is conformed of 38
model layers, variable height and terrain following, model levels, up to a pressure of 50 hPa. The first model layer has a mean
height of 59m over Delhi (and a mean height of 56m over the whole model domain).
Previous studies using the VBS have used scaling factors from POA to derive SVOC emissions in each volatility bin based on
equilibrium partitioning calculations, as well as volatility distributions based on laboratory studies and assumed oxygenation
and chemical reaction rates (Shrivastava et al., 2011;Fountoukis et al., 2014). To investigate the impact of these assumptions
on the model predictions, we have modified the model code so that the VBS emissions, the oxygenation rates and VBS reaction
rates, can be directly controlled via namelist options. The parameters which are perturbed in this way for this study are
described in more detail in Section 2.3.
The volatility distribution of open biomass burning emissions is taken from May et al. (2013), and multiplied by a scaling
factor of 3 (based on equilibrium partitioning calculations) to ensure reasonably similar condensed mass at emission as that
reported in the FINN 1.5 emission dataset. Similar calculations have been made in previous studies, giving roughly the same
scaling factor (Shrivastava et al., 2011;Fountoukis et al., 2014; Denier Van Der Gon et al., 2015;Ciarelli et al., 2017). Before
applying the scaling factor we assumed a ratio of matter mass to carbon mass of 1.4, dividing the emission inventory matter
mass by this to obtain the carbon mass. Within the model each VBS compound is stored as two variables, the oxygen part and
the 'non-oxygen' part. When adding the emissions we multiply the carbon mass by 1.17 to get the 'non-oxygen' mass (carbon,
plus other atoms), and by 0.08 to get the oxygen mass. These scaling factors were taken from Shrivastava et al. (2011). We
then apply the SVOC scaling factor, and volatility distribution, to give the final SVOC emission profile. The IVOC scaling
factor is applied to the same base emissions to get the IVOC emission profile. The volatility distribution for anthropogenic
emissions is also multiplied by a scaling factor of 3 for the same reasons as above. It is worth mentioning that the perturbed
space explored here is embedded in the parent VBS scheme that has been adopted. There have been a large number of
developments in, and variants of, the VBS aiming to address particular questions related to SOA formation at various levels
of complexity; for example, the mechanistic measurement-constrained radical 2D-VBS examining the role of ELVOC and
ULVOC in new particle formation; (Zhao et al., 2020; Zhao et al., 2021). In the current study, our implementation has been
developed from the VBS version available in the distribution version of WRF-Chem and our results should be interpreted in
the context of the structural capabilities and limitations therein. More information about the VBS distributions and parameter
space setup is in section S1 in the supplementary material.
Anthropogenic emissions are derived from the EDGAR-HTAP, SAFAR-India (CRI-v2R5 speciation) and NMVOC global
emission datasets, with NMVOC emissions speciated for the CRI-v2R5 chemical scheme, and applying diurnal activity cycles
to the emissions based on emission sectors in Europe (Olivier et al., 2003). We used these diurnal activity cycles (Figure S1
in supplement) as there were no data available for activity behavior in Delhi. Biogenic emissions are calculated online using
the MEGAN model (Guenther et al., 2006). Biomass burning emissions are taken from the FINNv1.5 global inventory
(Wiedinmyer et al., 2011).

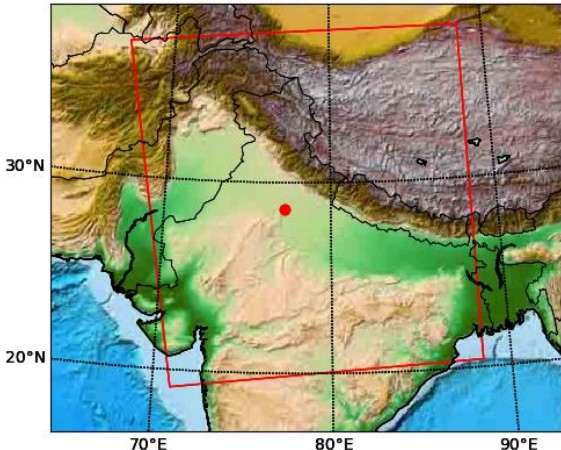

Figure 1. WRF-Chem model domain with topography data. The red marker highlights the location of IMD New Delhi,
where the AMS observations were taken and the red rectangle shows the area that covers the model results.
**2.2 Observations**
Aerosol observations were made at the Indian Meteorology Department (IMD) at Lodhi road in New Delhi, India (Lat 28.588,
Lon 77.217) from $26^{th}$ April to $30^{th}$ May 2018 as part of the PROMOTE campaign (Reyes-Villegas et al., 2021). A High-
Resolution Time-of-Flight Aerosol Mass Spectrometer (HR-TOF-AMS, Aerodyne Research Inc.), hereafter referred to as
AMS, was used to measure mass spectra of non-refractory particulate matter with an aerodynamic diameter equal or lower
than 1 µm ($PM_1$), including organic aerosols (OA), sulphate ($SO_4^{2-}$), nitrate ($NO_3^-$), ammonium ($NH_4^+$) and chloride ($Cl^-$), in a
5-minute time resolution. The AMS operation principle has been previously described by DeCarlo et al. (2006). The AMS was
calibrated during the campaign for the ionisation efficiency of nitrate (IE) and the relative ionisation efficiency (RIE) of other
inorganic compounds using nebulised ammonium nitrate and ammonium sulphate with a diameter of 300 nm. The data were
analysed using the IGOR Pro (WaveMetrics, Inc., Portland, OR, USA) based software SQUIRREL (Sequential Igor data
Retrieval) v.1.63I and PIKA (Peak Integration by Key Analysis) v.1.23I. The organic to carbon (O:C) ratios were calculated
with PIKA using the improved-Ambient elemental analysis method for AMS spectra measured in air (Canagaratna et al.,
2015). The AMS data, OA mass concentrations and O:C ratios, are used to compare with the WRF-Chem model outputs: total
organic matter mass concentration (Total_OM)  and organic to carbon ratios (OC_ratio).
There were no Planetary boundary layer height (PBLH) measurements available at IMD Lodhi road, hence, PBLH data were
sourced from ECMWF ERA5 with 0.25 deg. results in 1-hour resolution for the coordinates closest to the IMD site.
Meteorology data was downloaded from https://ncdc.noaa.gov/ (last access: 05/01/2019) for the Indira Gandhi International
Airport, India meteorology station.
The meteorology data were used to interpret the diurnal behaviour of the chemical species and to compare with meteorology
outputs from WRF-Chem. A dataset of meteorology was not available at IMD. The use of meteorology from airports has been
previously used and is considered to be representative of regional meteorology without being affected by surrounding buildings
(Reyes-Villegas et al., 2016).

**2.3 Perturbed Parameter Ensemble**

To evaluate the sensitivity to variations in the VBS emission and processing parameters of our WRF-Chem model of the
simulated OA over the New Delhi region, we generated a perturbed parameter ensemble (PPE). We choose a set of simulations
with optimal space-filling properties that provide effective coverage across the multi-dimensional space of the uncertain model
parameters. Here, we perturb ten parameters of the WRF-Chem model that relate to semi-volatile POA emissions and the aging
of these VBS compounds. The parameters correspond to five processes in the model, which are perturbed with respect to both
anthropogenic emissions and biomass burning emissions. These process parameters are:
1. **VBS ageing rate:** The reaction rates of VBS compounds with OH - each reaction reduces the volatility of the
compound by a factor of 10 (1 decade in saturation concentration, $Ci^*$, position), and adds between 7.5% and 40%
oxygen (determined by the SVOC oxidation rate parameter, below). $Ci^*$ is the condensed mass loading at which half
of the organic material in that volatility bin will be in the condensed phase, and half will be in gas phase (Donahue et
al., 2006).
2. **SVOC volatility distribution:** This parameter is expressed in terms of an "equivalent age", determined using a simple
ageing model. At time = 0 all VBS molecules will be highly volatile, with a $Ci^* = 4$. These compounds are processed
at a fixed reaction rate (at each step 0.1% of the gaseous mass in a volatility bin is moved to the next volatility bin),
with simple equilibrium partitioning of the VBS components between the gas and condensed phases (to roughly
simulate the manner in which VBS compounds are partitioned and aged within the WRF-Chem scheme). This
processing reduces the overall volatility of the VBS compounds, first providing a spread of mass across the volatility
range, before accumulating the mass in the lowest volatility bins until 90% of the VBS mass is in the $Ci^* = -2$ volatility
bin ("time" = 1). This parameter is a scalar variable (between 0-1), that indicates the dimensionless position between
these two points, and has an associated volatility distribution. After examining the range of volatility distributions
given by this simple ageing model, we have chosen to use distributions within the range of 0.05 to 0.4. Using values
above 0.05 ensures there will always be some lower volatility compounds to condense. Above 0.4 almost everything
is condensed, so we have excluded values above this so that our PPE does not become too heavily weighted towards
these scenarios.  The scalar variable represents a sensible range of possible emitted volatility distributions. A method
was needed by which we could represent the variation of possible volatility distributions within the process emulator.
The direct approach would be to include a scaling factor for each volatility bin as separate parameters. However, this
would have greatly increased the complexity and size of our parameter space, and these parameters would not be
independent of each other, leading to a lot of wasted parameter space, waste in the use of our limited computer
resources available for the PPE simulations and the assumptions for our variance-based sensitivity analysis becoming
invalid. Instead, we used a simple reaction model, where each step in a fraction of each volatility bin would be 'aged'
and moved to the next volatility bin. This approach also allowed us to include some simple partitioning, with aging
process stopped for any condensed matter; replicating the behaviour of the model these distributions will later be
injected into. Given that we used a simple, fractional, aging process, it would not be appropriate for us to try to relate
it to a physical variable. We have included Figure S2 instead, which gives example volatility distributions through
the range of this scalar value used in our study.
3. **SVOC oxidation rate:** This parameter represents the degree of oxidation that occurs with (or is induced by) each
reaction with an OH molecule. Previous studies have used values between 0.075 (7.5%) extra oxygen (or one oxygen
atom)  (Robinson et al., 2007) and 0.40 (40%) extra oxygen (or five extra oxygen atoms per reaction ) (Grieshop et

al., 2009). Grieshop et al. (2009) stated that with 7.5%, there is not enough addition of oxygen to the organic mass, while with the 40% there is a noticeable improvement to the OA oxygen content with little effect on the predicted organic mass production. In our study, the lowest level is 0.075 extra oxygen (or one oxygen atom) and the uppermost level is 0.45 (or six extra oxygen atoms per reaction).

4. **IVOC scaling:** IVOC compounds bridge the gap from SVOC to VOC ($\log10(C^*)$ 4-6). Including the IVOC independently to parameter (2) (based on our simple ageing model) enables us to still include these within the volatility distribution (this does restrict the impact of parameter (2) to influencing the shape of the volatility distribution for the lower $C^*$ values only. These IVOC emissions are calculated using a fixed volatility distribution which scales from the non-volatile OA mass in the emissions inventory. The fractional emitted masses are: 0.2 for $Ci^* = 4$; 0.5 for $Ci^* = 5$; and 0.8 for $Ci^* = 6$ (as shown in supplementary Figure S2) (0.2+0.5+0.8=1.5), this is the initial emission amount that then will be scaled by another factor, between 0-3, to probe the sensitivity of the model to the abundance of IVOCs.

5. **SVOC scaling:** This parameter is the scaling factor of the SVOC emissions, (which have been given a volatility distribution by parameter 2). Traditionally such scaling has been used: to ensure that the condensed mass of the emitted SVOC is the same as the non-volatile OA mass in the emissions inventory; however, this scaling could also be used to off-set errors in the emission inventory estimates of OA emissions. The scaling needed to ensure that the emitted condensed mass is the same will never be less than 1, but could go to x20 (or more) for the "younger" SVOC volatility ranges (as estimated using the equilibrium partitioning tool for parameter 2). However, in order to accommodate potential over-estimates of the emission inventories, and to avoid too much OA being generated after aging of any highly-volatile emissions, we chose an SVOC scaling range 0.5 to 4.

Table 2 shows the uncertainty ranges applied to each of the parameters, that we explore with the PPE, and Table S3 in the supplementary information shows an example of a 'namelist.input' file with the parameters to control the VBS scheme, that was used to create the model simulation. A total of 111 model simulations make up the ensemble. Following the statistical methodology outlined in Lee et al. (2011), the combinations of input parameters used for the simulations in the PPE were selected using an optimal Latin hypercube statistical design algorithm (Stocki, 2005), providing a good coverage of the multi-dimensional parameter space. The selection of combinations was performed in three subsets, for use in building statistical emulators to densely sample key outputs from the model over its uncertainties. First, a single design of 61 runs was generated for training the emulators (subset 1), and then a second set of 20 runs was made that 'augmented' into the larger gaps of the first design, for use in validating the emulators (subset 2). On an initial comparison to observations, the observations were found to be outside the range of the PPE's output, and following an investigation into this, the lower bound of the anthropogenic SVOC scaling parameter (parameter 5) was extended from 0.5x down to 0.1x. Hence, an extra, third, set of 30 runs were designed and simulated to cover the extended parameter space (subset 3), leading to a total of 111 runs in the final PPE. Table S4 in supplementary information provides a list of the model runs that make up the PPE with their respective values.

Table 2: Range of the parameter space used for SVOCs co-emitted within anthropogenic POA in the PPE with 111 model variants.

| Parameter number | Parameter name | min | Max |
|---|---|---|---|
| 1 | Anthropogenic VBS ageing rate ($cm^3$ $molec^{-1}$ $s^{-1}$) | 1.00E-13 | 1.00E-11 |
| 2 | Anthropogenic SVOC volatility distribution | 0.05 | 0.4 |
| 3 | Added oxygen per generation of ageing | 0.075 | 0.45 |
| 4 | Anthropogenic IVOC scaling | 0 | 3 |
| 5 | Anthropogenic SVOC scaling | * 0.1 | 4 |
| 6 | Biomass Burning VBS ageing rate ($cm^3$ $molec^{-1}$ $s^{-1}$) | 1.00E-13 | 1.00E-11 |

| 7 | Biomass Burning SVOC volatility distribution | 0.05 | 0.4 |
| 8 | Added oxygen per generation of ageing | 0.075 | 0.45 |
| 9 | Biomass Burning IVOC scaling | 0 | 3 |
| 10 | Biomass Burning SVOC scaling | 0.5 | 4 |

\* 81 runs were performed with an anthropogenic SVOC scaling min = 0.5 and max = 4 and 30 runs were performed with an anthropogenic SVOC scaling min = 0.1 and max = 0.5. This due to a min = 0.5 and max = 4 giving high Org mass concentrations, when compared with AMS.

**2.4 Emulation**

For each PPE member, a time series of the OC_ratio and Total_OM from the WRF-Chem model run was extracted at the closest coordinates to the IMD site (Lat 28.628, Lon 77.209) in the model output. Gaussian process emulators (O'Hagan, 2006; Lee et al, 2011) were built using the PPE. Similarly to the approach described in Johnson et al. (2018), initial emulators were constructed using only training simulations (subsets 1 and 3) and these were validated using the validation runs (subset 2). Once validated, a further new emulator was then constructed using both the training and validation simulations of the PPE together as training data, to obtain a final emulator based on all of the information that the PPE contains. An additional verification of the quality of each final emulator was obtained via a 'leave-one-out' validation procedure (where each simulation in turn is removed from the full set of 111 runs and a new emulator is built and used to predict that removed simulation).

Monte Carlo sampling of the emulators enabled dense samples of model output to be generated over the 10-dimensional parameter uncertainty of the model. We produced output samples for a set of 0.5 million input parameter combinations across the uncertainty space, hereafter called 'model variants', to explore the model's uncertainty.

**2.5 Model evaluation**

Alongside the emulation, outputs from the 111 model runs (OC_ratio and Total_OM) were additionally evaluated, against the AMS observations (O:C and OA), using various model evaluation tools, including the fraction of predictions within a factor of two (FAC2), mean bias (MB) and the index of agreement (IOA). Section S3 of the supplementary information provides a detailed explanation of the calculations for each evaluation metric and information on how to interpret the values.

**3 Results and discussion**

**3.1 Model outputs and observational analysis**

The model outputs of the central WRF-Chem run, from the original parameter space (Subsets 1 and 2), are used to compare with observations in order to analyse the model performance. As mentioned in the methods section, the VBS setup will directly affect OA concentrations and PM. The oxidative budget for inorganic chemistry is not directly affected, however, by changing the aerosol size distribution there are some indirect effects on inorganic aerosol and gaseous species through changes in aerosol water content, cloud fields, and aerosol-radiation interactions. Figure 2 shows the comparison for the full dataset (1st – 29th May 2018) between model outputs and observations performed at IMD Lodhi road, where we see higher $PM_{2.5}$ and $NO_x$ concentrations in the model simulation. The high NOx concentrations in the model seem to be related to high $NO_2$ concentrations as the NO concentrations are in line with the range of the observations of NO. Looking at the meteorological parameters, we can see similar temperatures and wind speeds between the model and observations, with lower RH and higher PBLH in the model.

306

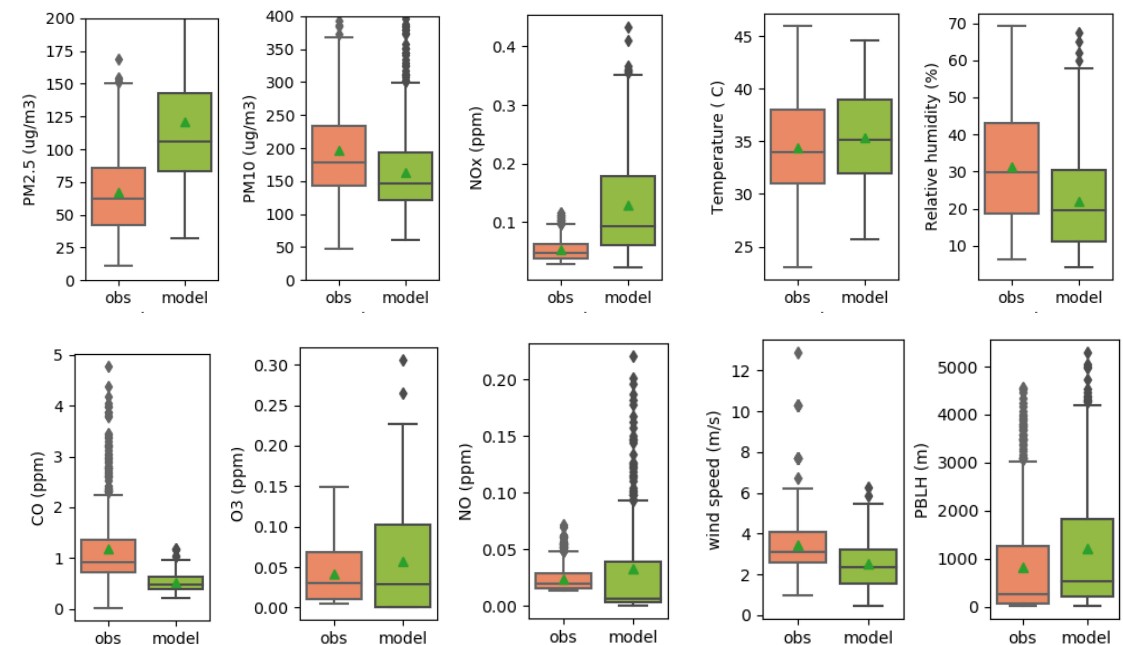

307

Figure 2. Comparison of observations (At Lodhi Road for air quality and IGI Airport for meteorology parameters) and model
outputs of various parameters. May 2018. Bars highlight medians, quartiles and 95%, triangles highlight the mean.

**3.2 Model runs and AMS observations**

Here, we analyse and compare the mean values of Total_OM (modelled particle phase) and OC_ratio for the full period, 1st –
29th May 2018, of the 111 WRF-Chem model runs (Table S4 in supplement) with the AMS observations (OA and O:C).  The
top panel in figure 3 shows a bar plot of the mean OC_ratio for the model runs coloured by the mean total_OM concentrations.
The bottom panel shows the mean total_OM concentrations for the model runs coloured by the mean OC_ratio. The model
runs are sorted from low to high values of the y-variable. The continuous and dashed red lines show the mean ± one standard
deviation (SD) of the O:C ratio (top) and OA (bottom) measured with the AMS.  In general, compared to mean values measured
with the AMS, a large number of WRF-Chem runs had a low O:C_ratio and high mean Total_OM concentrations. The bottom
panel shows the mean total_OM concentrations of 47 runs lay within one SD of the mean OA concentration of 21.77 µg.m$^{-3}$
measured with the AMS. Moreover, the model runs with mean Total_OM concentrations near the mean OA concentrations
have OC_ratio mean values near the O:C mean AMS value (0.5), with a cyan colour. We explored a range of emission
multipliers (both IVOC and SVOC scaling). These upper limits, which have been of an appropriate magnitude for previous
studies in other locations using different emission datasets (e.g. Shrivastava et al. (2011)), turned out to be too high for our
emission dataset, and these are the model runs which produced the very high OM mass loadings (rather than these being
predominately caused by high oxidation rates). When the OM mass loading is high, more of the higher volatility (and, here,
less aged) compounds condense into the condensed-phase. The VBS scheme we have used has only gas-phase reactions, and
so once in the condensed-phase these compounds do not age further. This process leads to the lower mean O:C ratios that are
observed here. This analysis shows a number of model runs with mean Total_OM and OC_ratio values near the mean values
measured with an AMS.

329
330
331

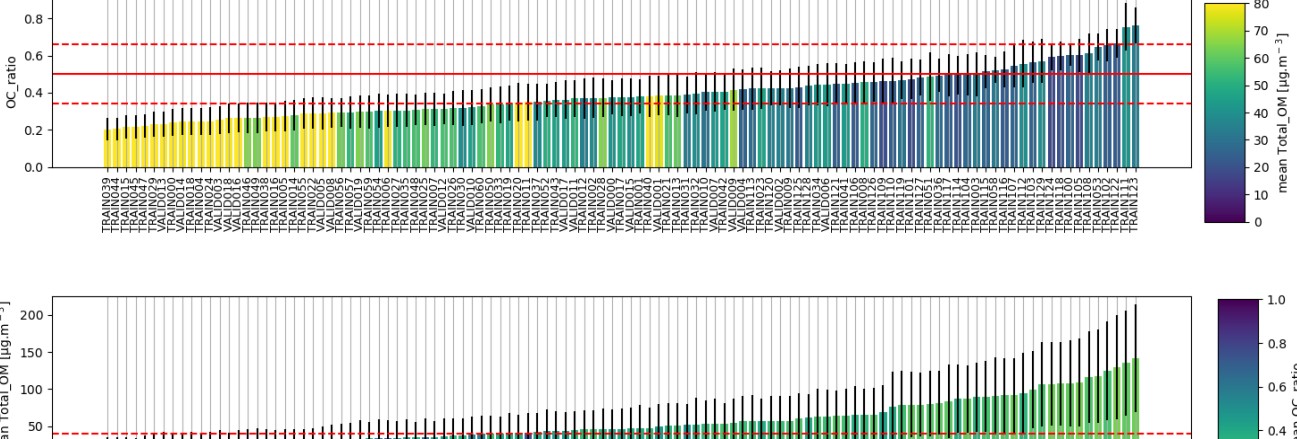

332

333

Figure 3. Analysis of the 111 model runs for the full period. Mean OC_ratio coloured by mean Total_OM (top plot) and mean Total_OM coloured by mean OC_ratio (bottom plot). The red lines highlight the mean ± SD of AMS observations (O:C top and OA bottom). The mean AMS values are O:C = 0.5 and OA = 21.77 µg.m$^{-3}$.

**3.3 Diurnal analysis to WRF-Chem runs**

The high time resolution data collected with the AMS provides the opportunity of analysing the WRF-Chem outputs in more detail, for example by looking at the diurnal cycles. Figure 4 shows the diurnal cycles of chosen WRF-Chem runs with Total_OM concentrations and OC_ratio close to the AMS observations. In the model runs, we were able to span high and low Total_OM and OC_ratio. However, in the case of OC_ratio, we were not able to span the range of the O:C from AMS observations with mean values of 0.3 at night and 0.7 during the day. Looking at the Total_OM concentrations, we identified two potential structural errors in the WRF-Chem outputs, the early morning peak and the late evening low concentrations. This could be due to application of unsuitable diurnal activity cycles to the emissions or WRF-Chem not being able to capture completely the dynamics of the planetary boundary layer. With no activity data available for Delhi, we used diurnal cycles of activities based on emission sectors in Europe (Olivier et al., 2003) (Figure S1 in supplement).We can observe in figure S5 a slightly better comparison in CO model vs observations, with flatter CO concentrations when looking at the observations.  For the diurnal cycles of meteorology (Figure S4), we can see that the model agrees with the PBLH- ERA5 in the early morning and until 14:00 h, time when PBLH- ERA5 starts dropping and PBLH-Model remains high, perhaps preventing concentrations to accumulate. This makes building and testing the emulator challenging as we may get the correct concentrations for the wrong reasons. The emulator can be built over a specific time-period and be compared with the observations. Hence, the emulator was built over two periods of interest; the full period (1$^{st}$-29$^{th}$ May) and a period where no potential structural errors were identified from 14:00- 16:00 hrs for 1$^{st}$-29$^{th}$ May (2-4 pm period). Emulator analysis involving the filtering of model results to avoid structural errors has been successfully performed previously in constraining a climate model (Johnson et al., 2020). Looking at the mean OC_ratio and Total_OM of the model runs for the 2_4 pm period (Figure S6), 34 runs lay within one SD of the OA mean concentration (12.20 µg.m$^{-3}$) measured with the AMS, compared with the 47 runs identified from figure 3. This means that even by analysing the 2-4 pm period we still have model runs that cover the AMS observations.


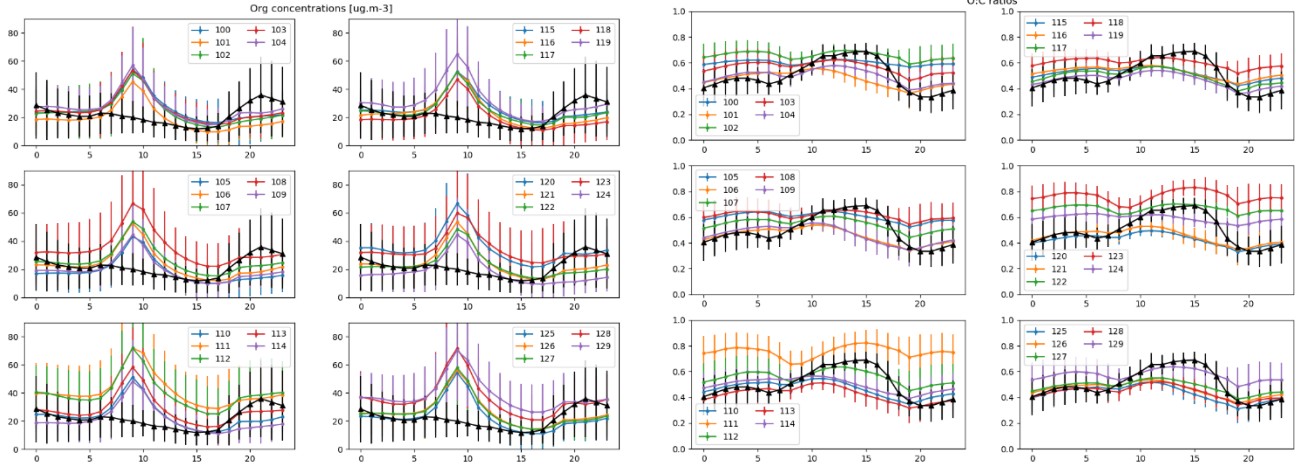


Figure 4. Diurnal cycle of selected WRF-Chem runs with values near the AMS observations (black line).
**3.4 Model evaluation**
There are various tools that can be used to compare the model outputs with the observations. In this study, we use a number
of statistical metrics (see Section S3 in the supplementary information for a detailed description of each metric we consider)
to evaluate the ensemble of 111 model runs for the 2-4 pm period and the full period. The fraction of predictions within a
factor of two (FAC2) represents the fraction of data where predictions are within a factor of two of observations. The Mean
Bias (MB) gives an indication of the mean over- or under-estimation of predictions. The Index of Agreement (IOA) is a
commonly used metric in model evaluation (Willmott et al., 2012), ranging between -1 and +1, with values close to +1
representing a better model performance. Table S3 shows the results of the model evaluation for the 2-4 pm period and table
S4 the results for the full period. When comparing the performance of the two periods; the model runs of the 2-4 pm period
have a better performance with 103 runs for O:C and 29 runs for OA with FAC2 > 0.6 compared to 94 runs for O:C and 4 runs
for OA with FAC2 > 0.6 for the full period. The negative MB in O:C suggests the models are underestimating the O:C ratios
(between -0.01 to -0.15) measured with the AMS. However, the FAC2 values of 0.96 and higher indicate that the models are
doing a good job overall at simulating the O:C ratios. This is not the same for OA concentrations, where the models show an
over-estimate of the concentration compared to observations, and where only 0.56 -0.62 of predictions were within a factor of
two of the OA observations.
The IOA provides similar results with a better model performance in the 2-4 pm period, with 10 model runs for the 2-4 pm
period and only two runs for the full period with IOA values equal or higher than 0.45. It is interesting to see that while FAC2
was higher, for OA and O:C, in the 2-4 period runs compared to the full period, IOA values in 2-4 period were high with OA
but low with O:C, which reached IOA values of 0.53 in the 2-4 period and 0.56 in the full period. Previous studies performing
modelling evaluation determined similar IOA values using various models (Ciarelli et al., 2017;Fanourgakis et al., 2019) . For
instance, Chen et al. (2021), modelling SOA formation, obtained IOA between 0.39 – 0.49. Huang et al. (2021) published
recommendations on model evaluation and identified IOA of around 0.5 for organic carbon. Lee et al. (2020) performed a
sensitivity analysis to two different SOA modules and obtained IOA values of 0.46 – 0.52.
The model evaluation metrics, along with the parameter setup for each ensemble member, allow us to analyse the model setup
that gives a better performance. Figure 5 shows the relative variation (%) of the five anthropogenic parameters of the PPE (1
– 5) for the 2-4 pm period (Figure S7 in supplementary material shows the analysis for the full period). Each pentagon
represents the 5-D parameter space and the positions of the dots connected with lines show the position of each parameter
within its range for that specific ensemble member. The filled area within the dots represents the explored parameter space in
each ensemble member. We are analysing the five anthropogenic PPE only since the five parameters related to biomass burning
represented a low contribution to the Total_OM concentrations. We are looking for blue, light blue or green colours in the
lines and dots (high FAC2 values from the O:C analysis) and blue, light blue or green colours in the filled area (high FAC2
values from the OA analysis) to identify the model runs with a good evaluation. In figure 5, we can see that the best runs
according to the O:C and OA model evaluation are TRAIN127 and TRAIN121 with other TRAIN runs also with good
performance such as (126, 036, 117 ,104, 115, 119 and 058). In general, these model runs have low SVOC volatility distribution
(emitted VBS compounds are more volatile) and SVOC scaling. TRAIN127 and TRAIN121 have low VBS ageing rate, SVOC
volatility distribution and SVOC scaling and with either high SVOC Oxidation rate or high IVOC scaling.

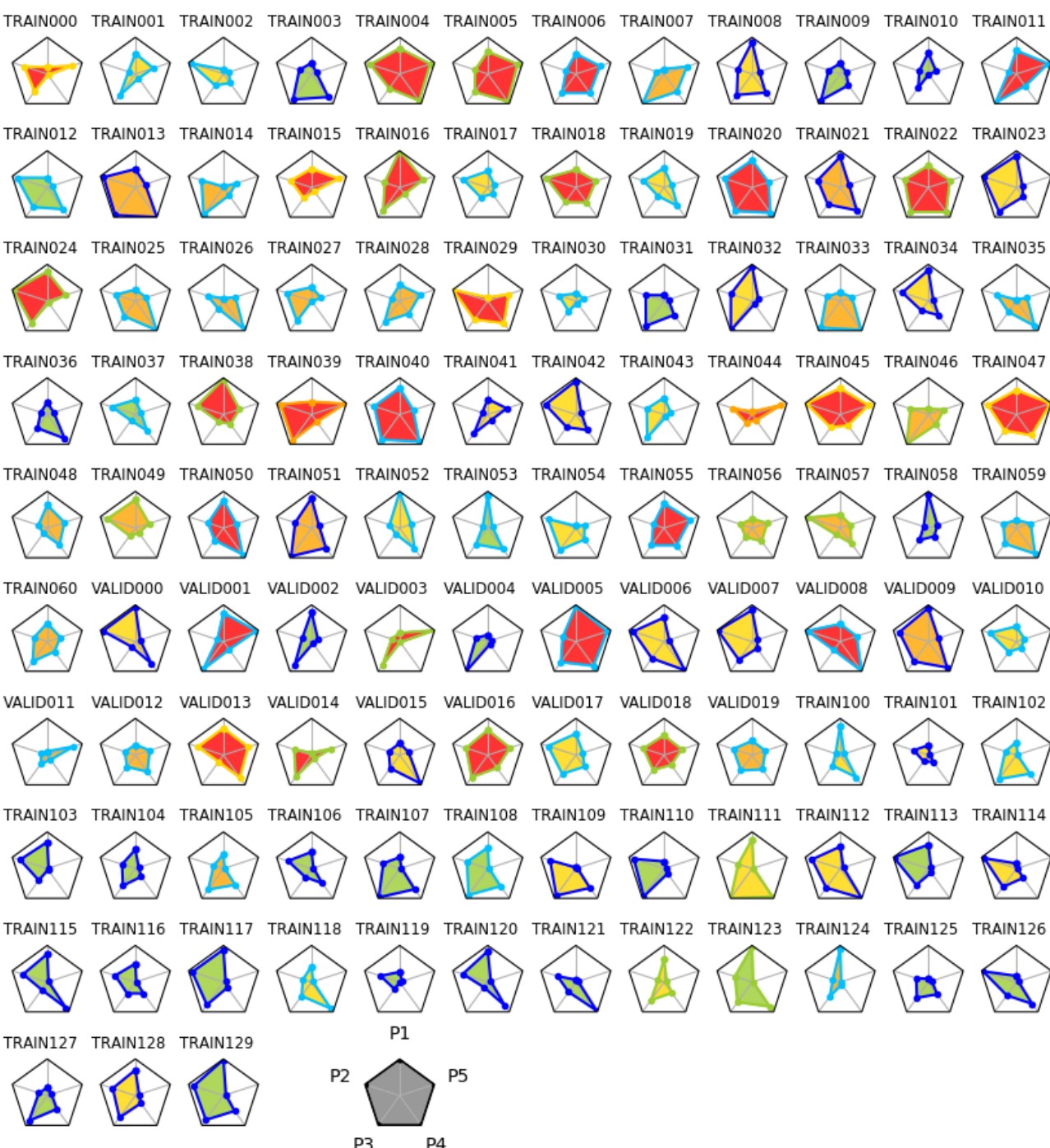

Figure 5. Relative variation (%) of the 5 anthropogenic PPE (1 – 5) for the 2-4 pm period. Each pentagon represents the 5-D parameter space and the positions of the dots connected with lines show the position of each parameter within its range for that specific ensemble member. The filled area within the dots represents the explored parameter space in each ensemble member. Anticlockwise from top there are the five anthropogenic parameters: VBS ageing rate (P1), SVOC volatility distribution (P2), SVOC Oxidation rate (P3), IVOC scaling (P4) and SVOC scaling (P5). The values of the 5 parameters have been normalised dividing by their respective maximum values, hence their values in this plot range from 0 – 1. The colour in the lines and dots represents the FAC2 values from the O:C analysis and the fill colour represents the FAC2 values from the OA analysis. Red = 0 – 0.2, orange = 0.2 – 0.4, yellow = 0.4 – 0.6, green = 0.6 – 0.8, light blue/cyan = 0.8 -0.9 and blue = 0.9 -1.0

## 3.5 Emulator analysis

### 3.5.1 Emulator building and testing

Once we confirm that the ensemble of 111 model runs span the AMS observations we can use it to build the emulator. The emulators are tested using the leave-one-out validation approach (Johnson et al., 2018). In this analysis, each ensemble run is first excluded from the emulator build, and then the emulator is used to predict the output at the parameter setting of the excluded run. Figure 6 shows plots of the emulator predictions (with 95% credible intervals from the emulator model) vs the model outputs of the 111 runs from the leave-one-out validation for OA. Predictions from a perfect emulator would follow exactly along the 1:1 line on the plots.

We built and tested the emulator for the full period ($1^{st}$ – $29^{th}$ May) to have an overview of the emulator performance. The emulator can be built over a specific time-period to compare with the observations. This allows to study the model performance under different conditions, i.e., high/low aerosol concentrations, day/night, etc. We selected four period time-slots to build and test the emulator under high and low Total_OM concentrations and two time-slots. These four emulators showed a good validation analysis (Refer to section S5.1.1 in the supplementary material). However, due to the potential structural errors identified from the diurnal analysis (Section 3.3), we will focus on the selected period without structural errors, 2-4 pm period. Figures S11 and S12 in supplementary material show the spread of Total_OM and OC_ratio respectively, for the ensemble of 111 model runs vs the 10 parameters.

We see in Figure 6 that overall, the emulators built for the two periods; full period (6.a and 6.b) and 2-4 pm period (6.c and 6.d) show a good performance; For the 2-4 pm period, Total_OM with only nine runs that are not within the 95% CI from prediction (red markers) and OC_ratio with ten runs that are not within the 95% CI from prediction. With the new 30 runs (error bars in blue) we managed to reduce the Total_OM concentrations with good prediction on the emulator. However, there is a compromise in the OC_ratio with eight runs with high OC_ratio values that at not within the 95% of the prediction interval of the emulator.

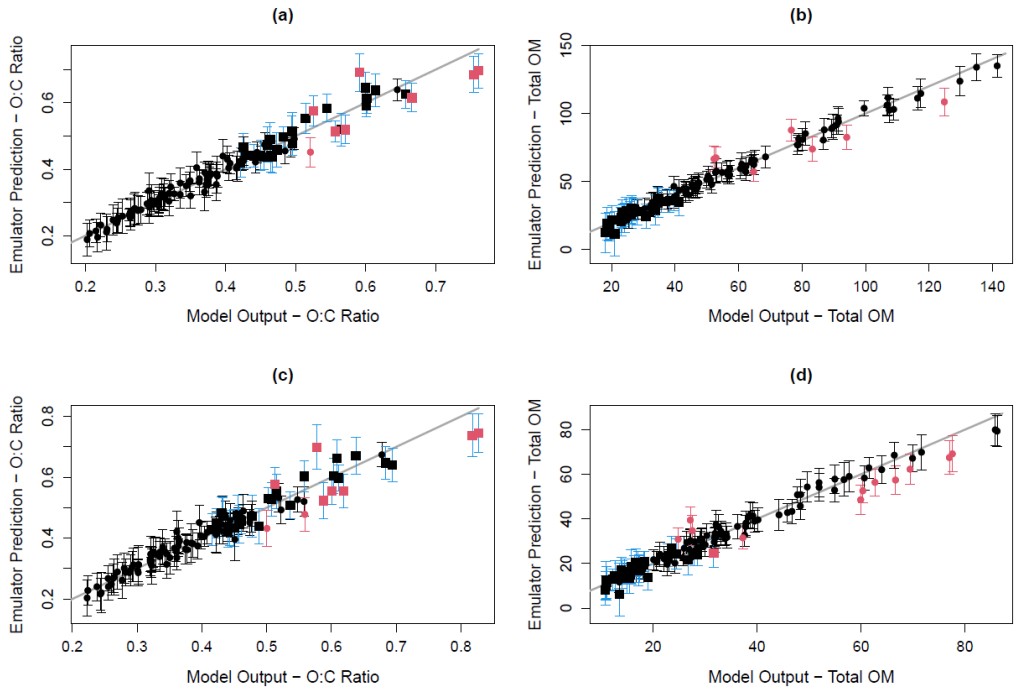

431

432

Figure 6. Validation of the full (a and b) and 2-4 pm (c and d) periods for O:C ratio and Total OM. Circles are the original 81 runs. Squares with error bars in blue are the new 30 runs with low settings of the anthropogenic SVOC scaling parameter (which has led to low aerosol mass). Runs where the actual model output lies outside the 95% prediction interval of the emulator are shown in red.

**3.5.2 Emulator sensitivity analysis**

We use a variance-based sensitivity analysis (Lee et al., 2011;Johnson et al., 2018) to decompose the overall variance in the model output for key variables of interest into percentage fractions for the 10 parameters. This analysis was performed to the full period and the 2-4 pm period (Figure 7). Looking at the parameters for the two periods, the anthropogenic SVOC scaling has the highest contribution to the variance, which suggests that constraining this parameter would lead to a reduction in the uncertainty in these outputs from the model. Anthropogenic SVOC volatility distribution has some impact on O:C ratios with a fraction of variance of around 15%.

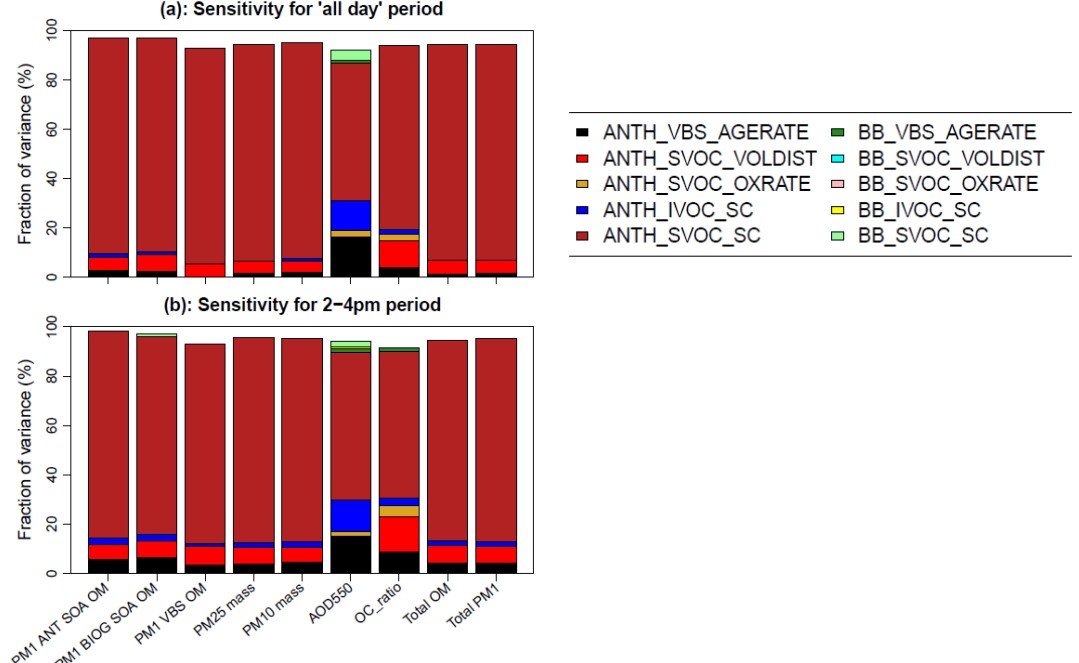



Figure 7. Sensitivity evaluation of the 10 chosen parameters for the 2-4 pm period (a) and the full period (b).
**3.5.3 Impact of constraint on uncertainty**
The emulator was used to predict model outputs for a sample of size 0.5 million, for the full period and the 2-4 pm period.
Figure 8 shows the probability distribution of OC_ ratio and Total_OM predicted over the full parameter uncertainty. The
AMS mean ± 1SD are shown in red. We can see the higher density (lower values) of the Total_OM show a good agreement
with the AMS-OA concentrations. However, in the case of O:C, the higher density lies on the low O:C ratios compared to the
O:C-AMS observations which lie in the upper tail of the predicted distribution. The OC_ratio varies within the two periods,
with a wider density range for the full period, 0.25-0.55, which represents the variability of the OC_ratio over the full day. In
the case of the 2-4 pm period, we can see more narrow density, 0.3-0.5, which, while lower than the mean O:C ratio measured
with the AMS (0.65), may be representative of the O:C ratios estimated with the WRF-Chem runs. This suggests that when
analysing diurnal behaviour of WRF-Chem outputs without structural errors, we would be able to analyse more into detail the
WRF-Chem performance over different hours of the day.

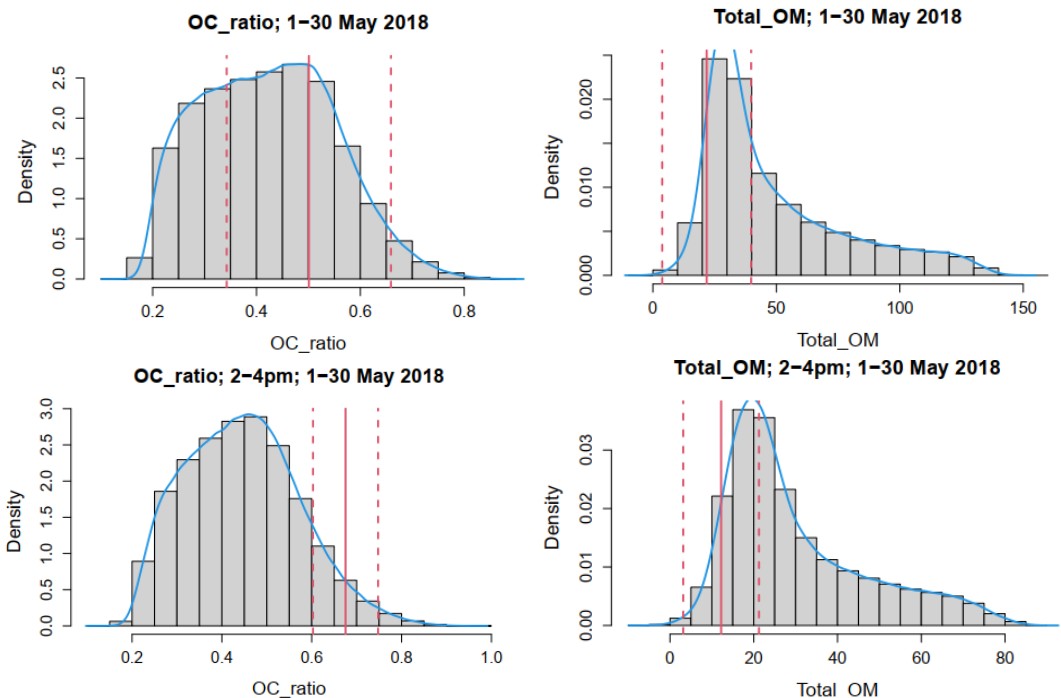

Figure 8. 0.5 million emulator sample, before constraint, covering the full parameter uncertainty space of the model for the full period (a and b) and for the 2-4 pm period (c and d). Red highlights the AMS mean +/- SD observations.

**3.5.4 Constraint effect.**

The AMS observations, OA concentrations and O:C ratios, are used to constrain the emulation, applying an observation uncertainty as mean ± SD. With mean as the emulator prediction and 1 SD uncertainty, we apply the constraint when accounting for emulator prediction uncertainty, by retaining the variant if the range mean ± SD overlaps with the observation uncertainty range.

Figure 9 is a 2-d histogram for joint constraint (Total_OM and OC_ratio) for the 2-4pm period, with colour showing frequency of variants in a pixel of an underlying grid arranged as a pairwise (shown by the label box on each axis (above/to right). Each 2-d pairwise space has been split into a 25x25 uniform grid to calculate the frequencies. Where the plots show yellow to red, more variants are retained than in the green / blue areas, highlighting the most likely (higher probability) area of space. This analysis shows that when constraining both Total_OM and O:C ratios, the emulator retains 52310 variants from 0.5 million, which is approximately a 10.46% of the original variants generated. Figure S13 shows the histogram

White areas indicate no variants at all retained in that pixel, so that 2-d space is ruled out with respect to all 10 dimensions. (probability=0). Where the colour is uniform, e.g., biomass burning parameter plots in figure 9, the parameter is essentially un-constrained, and all parts of parameter space with respect to those 2 parameters are equally likely/covered by variants (as it was before the constraint was applied). These plots show where in parameter space is most likely given the comparison to observation. These are the variants that we cannot rule out (are plausible) given the uncertainty – it does not mean they are all 'good'. It is worth mentioning that with this analysis we do not locate the exact 'best' run, we provide a range of potential combinations to test the WRF-Chem set-up.

These results agree with the analysis in the model evaluation (Section 3.4). Figure 9 shows, in red colour, the higher probability
that with low SVOC volatility distribution and low SVOC scaling would give a good model performance. However, there is
no clear pattern with the other parameters.

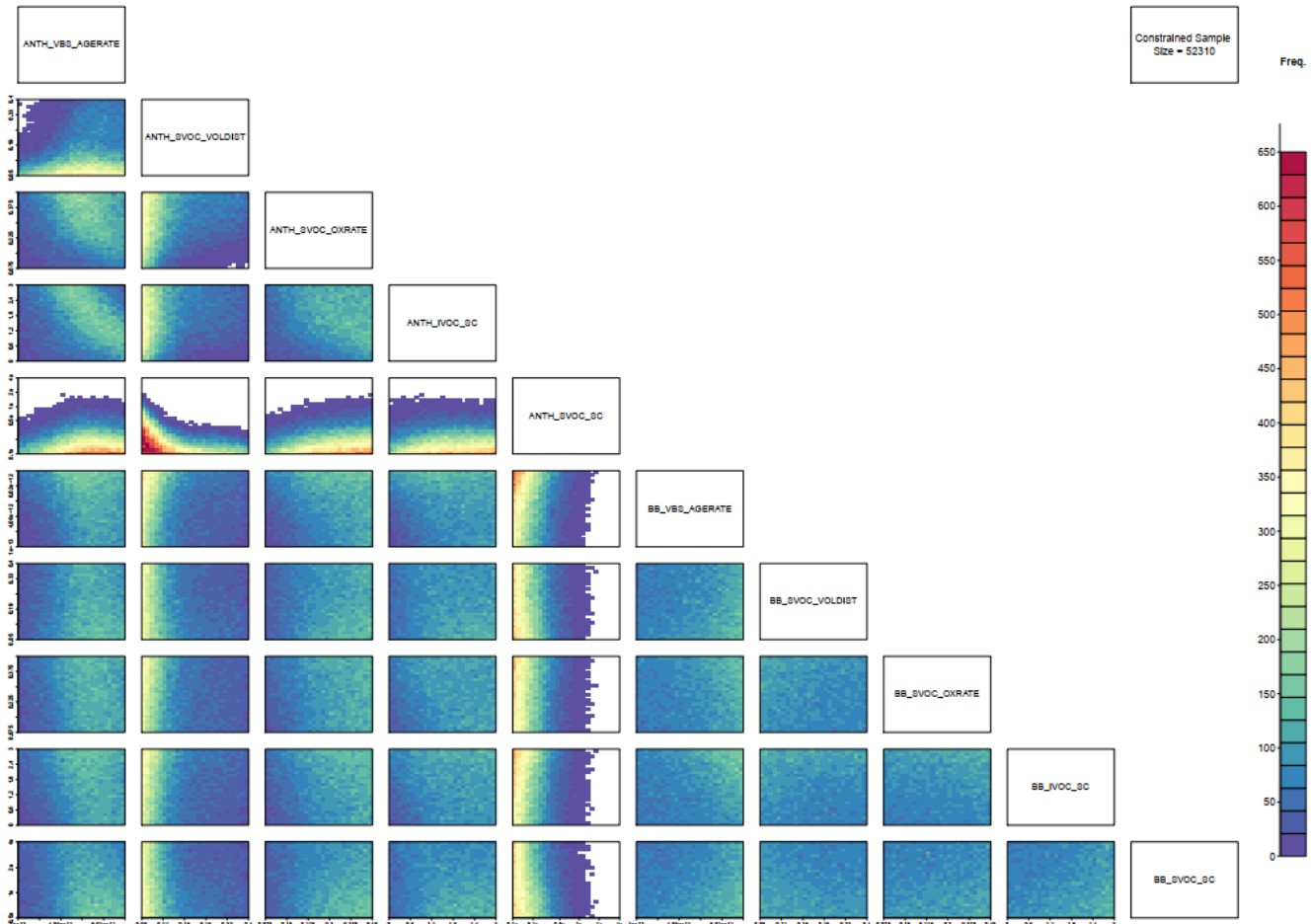


Figure 9. 2-d histogram for joint constraint effect (Total_OM and OC_ratio) accounting for emulator uncertainty. Retain
52310 variants from 0.5 million emulations (~10.46%).

### 3.5.5 Marginal parameter constraints.

Figure 10 shows the marginal constraint (1-d projection) on the parameters over their ranges. The unconstrained sample (black)
has even coverage (is sampled uniformly) across all parameter ranges and the parameter space. The unconstrained sample
covers the full 10-d space.
Where the probability density function (pdf) of the constrained sample is above the black unconstrained pdf, this means the
likelihood of the parameter taking a value at that point of its range is increased on constraint (more probability). Where it is

below, it is now less likely on constraint. (less probability). The more 'squashed' the unconstrained distribution is – the more the likelihood of the parameter taking values in the range with higher density is. This analysis is a useful tool to identify the more likely values of the 10 parameters over all the parameter space. Here, we can see that low SVOC volatility distribution and low SVOC are clear parameter values that we can use to improve the WRF-Chem model setup. Other parameters that we can start testing on WRF-Chem are; high BB VBS ageing rate (6) and BB IVOC scaling (9). It is worth highlighting the similarity of the effects on the anthropogenic and biomass burning parameters.

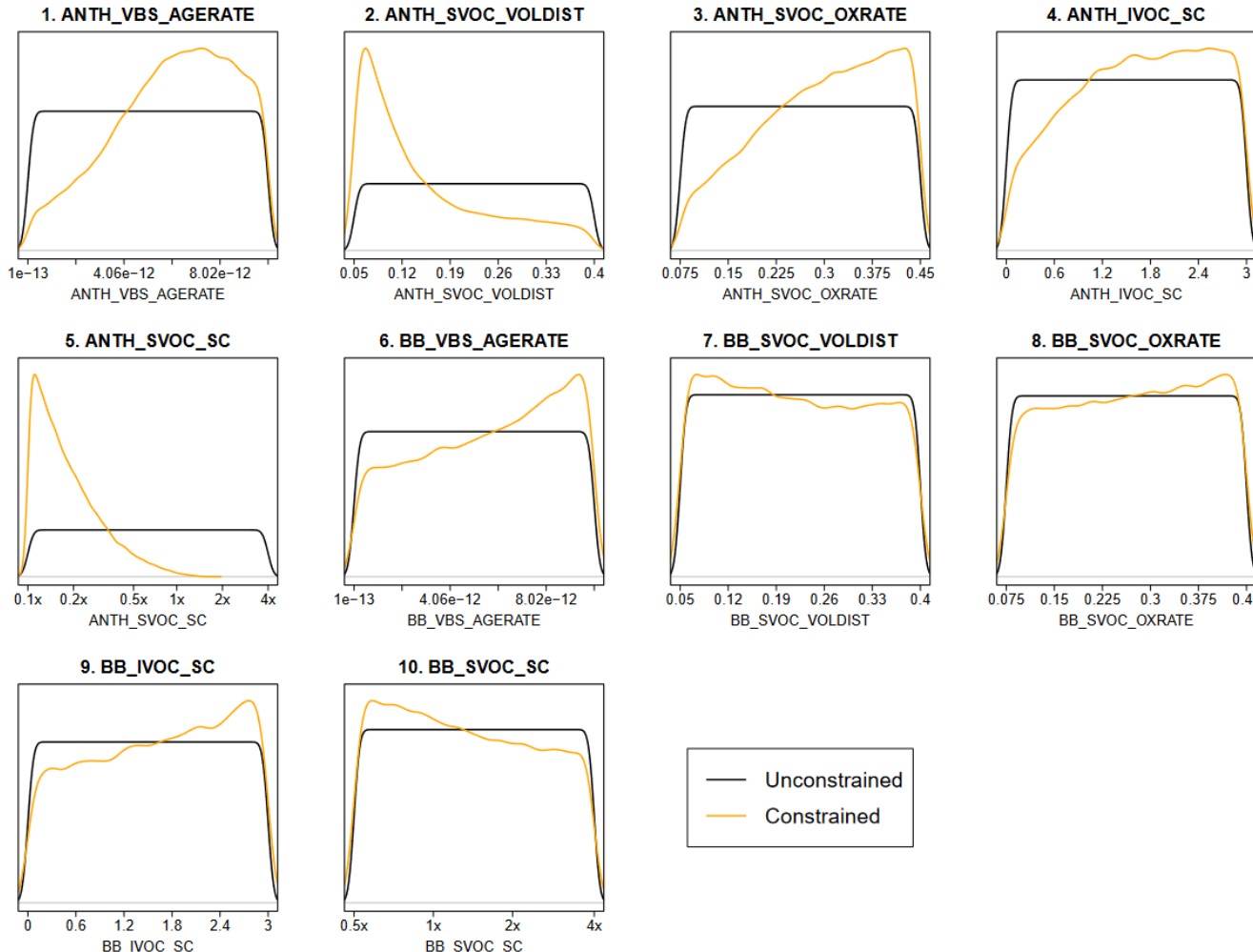

Figure 10. Marginal Parameter Constraints: joint constraint effect (Total_OM and OC_ratio).

## 3.6 Analysis of model evaluation and emulator runs.

Table 4 shows the WRF-Chem runs with both mean Org and mean O:C values close to AMS observations for the two periods and also selected runs from the 2-d histograms (Figure 9). Here we can see a couple of interesting findings. First, the O:C ratios presented a better performance with the model evaluation metrics; FAC2 values higher than 0.9 compared with FAC2 values up to 0.73 for the Total OM. Looking at the Total OM, there are higher FAC2 values in the 2-4 pm period, which might be related to the structural errors impacting the model performance int eh full period. The MB provides an estimation of the over prediction of the Total_OM. In this study, WRF-Chem runs were in general overpredicting the Total OM concentrations. Hence, MB is an important metric. In both periods, there are runs where the overprediction was 5 $\mu g.m^{-3}$ or lower, i.e, TRAIN110, TRANI121, TRAIN117, etc. This highlights the use of all the analysis presented in this study where we are able to identify probable values for the VBS model parameters and be able to model Total OM and O:C ratios.

Table 3. Analysis of model evaluation metrics and comparison with observations for the full and 2-4 pm periods. The FAC2 ranking is based on high FAC2 values of the Total_OM analysis. Mean AMS values for the full period: OA = 21.77 $\mu g.m^{-3}$ and O:C = 0.5. Mean AMS values for 2-4 pm period: OA = 12.20 $\mu g.m^{-3}$ and O:C = 0.67.

| Full period | Total_OM | | | | | | O:C ratio | | | | | |
|---|---|---|---|---|---|---|---|---|---|---|---|---|
| model | FAC2 ranking | FAC2 | MB | IOA | Total_OM mean | Total_OM SD | FAC2 ranking | FAC2 | MB | IOA | O:C ratio mean | O:C ratio SD |
| TRAIN110 | 1 | 0.62 | 2.23 | 0.45 | 23.75 | 16.58 | 27 | 0.94 | -0.04 | 0.48 | 0.46 | 0.12 |
| TRAIN126 | 2 | 0.61 | 5.13 | 0.38 | 26.42 | 19.83 | 20 | 0.95 | -0.04 | 0.51 | 0.46 | 0.11 |
| TRAIN119 | 5 | 0.60 | 9.54 | 0.31 | 30.83 | 22.05 | 7 | 0.97 | -0.04 | 0.54 | 0.47 | 0.10 |
| TRAIN117 | 6 | 0.59 | 3.18 | 0.41 | 24.56 | 16.93 | 10 | 0.97 | -0.01 | 0.53 | 0.49 | 0.11 |
| TRAIN009 | 8 | 0.59 | 10.54 | 0.30 | 68.50 | 36.13 | 15 | 0.96 | -0.08 | 0.51 | 0.42 | 0.11 |
| TRAIN121 | 9 | 0.59 | 2.87 | 0.41 | 24.17 | 18.59 | 21 | 0.95 | -0.05 | 0.50 | 0.45 | 0.11 |
| TRAIN104 | 11 | 0.58 | 5.77 | 0.39 | 24.17 | 18.59 | 8 | 0.97 | -0.01 | 0.56 | 0.45 | 0.11 |
| VALID002 | 12 | 0.58 | 13.27 | 0.24 | 34.49 | 24.15 | 2 | 0.98 | -0.08 | 0.52 | 0.43 | 0.09 |
| TRAIN003 | 13 | 0.57 | 12.65 | 0.24 | 33.73 | 23.56 | 6 | 0.97 | 0.00 | 0.55 | 0.50 | 0.12 |
| TRAIN127 | 16 | 0.56 | 4.78 | 0.37 | 26.12 | 20.02 | 5 | 0.97 | -0.02 | 0.55 | 0.48 | 0.10 |
| **2-4 pm period** | Total_OM | | | | | | O:C ratio | | | | | |
| model | FAC2 ranking | FAC2 | MB | IOA | Total_OM mean | Total_OM SD | FAC2 ranking | FAC2 | MB | IOA | O:C ratio mean | O:C ratio SD |
| TRAIN127 | 1 | 0.73 | 4.37 | 0.44 | 15.64 | 10.72 | 3 | 0.99 | 0.02 | 0.51 | 0.50 | 0.06 |
| TRAIN121 | 3 | 0.72 | 1.02 | 0.48 | 14.48 | 11.67 | 7 | 0.98 | 0.00 | 0.52 | 0.44 | 0.08 |
| TRAIN126 | 4 | 0.72 | 4.35 | 0.43 | 15.77 | 9.35 | 12 | 0.98 | 0.01 | 0.50 | 0.46 | 0.08 |
| TRAIN110 | 5 | 0.70 | 2.03 | 0.53 | 13.45 | 9.42 | 23 | 0.96 | 0.02 | 0.47 | 0.45 | 0.09 |
| TRAIN036 | 11 | 0.69 | 5.13 | 0.40 | 17.23 | 12.85 | 1 | 1.00 | 0.03 | 0.51 | 0.52 | 0.05 |
| TRAIN117 | 13 | 0.68 | 1.27 | 0.47 | 16.66 | 14.80 | 5 | 0.99 | 0.04 | 0.48 | 0.51 | 0.08 |
| TRAIN104 | 14 | 0.68 | 5.50 | 0.47 | 16.41 | 11.18 | 14 | 0.98 | 0.03 | 0.47 | 0.54 | 0.06 |
| TRAIN115 | 16 | 0.68 | 3.27 | 0.39 | 18.17 | 15.48 | 6 | 0.99 | 0.05 | 0.46 | 0.51 | 0.06 |
| TRAIN119 | 19 | 0.67 | 7.12 | 0.35 | 18.96 | 11.26 | 10 | 0.98 | 0.01 | 0.51 | 0.49 | 0.07 |
| TRAIN058 | 20 | 0.67 | 8.52 | 0.33 | 22.12 | 19.15 | 13 | 0.98 | 0.05 | 0.48 | 0.56 | 0.04 |

## 5 Conclusions

In this study we aimed to determine an effective way of tuning the VBS scheme using observations, and also to learn about the processes controlling OA in Delhi. WRF-Chem model runs with the VBS setup that successfully span the OA concentrations and O:C ratios from AMS observations can be identified, with many model runs overestimating organic mass concentrations and underestimating the O:C ratios compared with AMS observations. However, we identified two structural errors in the model related to a combination of unsuitable diurnal activity cycles applied to the emissions and/or WRF-Chem

not being able to capture completely the dynamics of the planetary boundary layer. It is worth mentioning that these structural errors might also be related to representation of other organic aerosol processes not represented by the VBS approach. As mentioned early in the introduction, this study only considers semi-volatile POA processes, without accounting for perturbations in SOA parameters and deposition processes. Recent studies, for example, have examined particle-phase and multiphase chemistry in aqueous aerosols and clouds (Shrivastava et al., 2022), and reactions of SOA precursors with other radicals like chlorine relevant to Indian conditions (Gunthe et al., 2021). Future studies could be focused on studying other parameters such as deposition processes and the perturbations in SOA parameters.

The structural errors prevented us from providing an optimised VBS approach in WRF-Chem. However, we were able to apply the emulator in two periods: the full period (1st -29th May) and the 2-4 pm period (14:00- 16:00 hrs, 1st-29th May) to present a methodology to evaluate a model performance using Gaussian emulators and metrics such as FAC2, IOA and MB. Optimization is a stage-by-stage process, future analysis would imply to do an emulation study to address diurnal activity and PBL directly, perhaps using NOx or total PM.

The performance of the two emulators, the full period and the 2-4 pm period, was similar, with the two emulators performing a good prediction of the model outputs and presenting a similar high variance of the anthropogenic SVOC scaling (Parameter 5). The model performance would highly improve if we are able to constrain the input values for the parameter 5.

When looking at the emulator sensibility analysis, we identified that the parameter anthropogenic SVOC scaling has the highest contribution to the variance, with fractions higher than 70%. This suggests that constraining this parameter would lead to a reduction in the uncertainty in these outputs from the model. Anthropogenic SVOC volatility distribution has little impact on the fraction of variance to O:C ratios with a fraction of variance of around 15%. None of the parameters show a clear variance to improve the model performance.

The model evaluation analysis based on FAC2, IOA and MB agreed with the emulator analysis in identifying that using low SVOC volatility distribution and low SVOC scaling would give improved model performance. Based on the MB analysis, for both the full and the 2-4 pm periods, there are runs where the Total OM overprediction was 5 ug.m$^{-3}$ or lower, i.e, TRAIN110, TRANI121, TRAIN117, etc. This overprediction is considered low compared to the mean Total_OM concentrations of ~20 – 30 µg.m$^{-3}$. Hence, we are able to identify probable values for the VBS model parameters and are able to model Total OM and O:C ratios in the range of the AMS observations.

The combination of the emulator analysis and the model evaluation metrics (FAC2, IOA and mean bias) allowed us to identify the plausible parameter combinations for the analysed periods. The more plausible combinations were found to be with a low SVOC volatility distribution and low SVOC scaling, which means a more volatile distribution. The methodology presented in this study is shown to be a useful approach to determine the model uncertainty and determine the optimal parameterisation to the WRF-Chem VBS setup. This information is valuable to increase our understanding on secondary organic aerosol formation, which in turn will help to improve regional and global model simulations, emission inventories as well as making informed decisions towards the improvement of air quality in urban environments.

**Data availability**

Emission generation scripts: https://github.com/douglowe/WRF_UoM_EMIT
Scripts for running WRF-Chem (and reducing the outputs to key diagnostics):
https://github.com/douglowe/promote_wrfchem_scripts

Scenario configuration files, and python script for calculating the "pseudo-age" of the emitted VBS:
https://github.com/douglowe/PROMOTE_VBS_scenarios
Scenario chemistry input files



**Financial support**

This research has been supported by the UK NERC and MoES, India through the PROMOTE project
under the Newton Bhabha Fund programme "Air Pollution and Human Health in a Developing
Megacity (APHH-India)", NERC grant numbers NE/P016480/1 and, NE/P016405. M.S. was supported
by the U.S. Department of Energy (DOE) Office of Science, Office of Biological and Environmental
Research (BER) through the Early Career Research Program. R.A.Z. acknowledges support from the
Office of Science of the U.S. DOE through the Atmospheric System Research (ASR) program at Pacific
Northwest National Laboratory (PNNL). PNNL is operated for DOE by Battelle Memorial Institute
under contract DE-AC06-76RLO 1830. This paper is based on interpretation of scientific results and in
no way reflects the viewpoint of the funding agencies.
Acknowledgements
We acknowledge use of the WRF-Chem preprocessor tools mozbc, fire_emiss, bio_emiss and anthro_emiss, provided by the
Atmospheric Chemistry Observations and Modeling Lab (ACOM) of NCAR.

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
