# Peer review of "approach: Exploring model uncertainty with a Gaussian Process emulator"

_Atmospheric Chemistry and Physics, 2022_

## Referee Comment (RC1)

**Review of: Simulating organic aerosol in Delhi with WRF-Chem using the VBS approach: Exploring model uncertainty with a Gaussian Process emulator.**

Reyes-Villegas et al., presents a modelling study with the WRF-Chem regional chemical transport model aiming at exploring the uncertainties of several parameters of the volatility basis set (VBS).
Up to 111 different model runs are performed with different SVOC and IVOC scaling factors, SVOC volatility distributions, as well as oxidation and aging rates using a 15km domain centred over New Delhi. Model results are compared against aerosol mass spectrometer (AMS) measurements performed during May 2018 in New Delhi. Additionally, the authors used the WRF-Chem model output as an input for Gaussian process emulators to explore the sources of model uncertainties.

The results indicated that the anthropogenic SVOC scaling factors, applied on top of the EDGAR-HTAP emission inventory (among others), had the highest contribution to the variance between the different model runs, with better agreements for low VBS ageing rate, low SVOC volatility distribution and low SVOC scaling factors combined with either high SVOC oxidation rate or high IVOC scaling factors.

The analysis of the model diurnal profiles against AMS measurements reveal several structural errors, with OA concentrations overestimated up to about a factor of 3 during morning peak hours. The authors conclude that deficiency in the model capability of correctly reproducing meteorological parameters, such as the dynamics of the planetary boundary layer, are likely the cause of these structural errors.

I find the study to be innovative, well designed and potentially useful for the atmospheric modelling community. The large amount of chemical transport model (CTM) runs (even though at coarse resolution) allow exploring the wide range of several physical and chemical parameters that are usually probed by the means of box-model simulations applied in more confined and constrained environments. Therefore, the comparison of such large amount of perturbed model runs with ambient AMS data, especially at hourly resolution, is novel and add significant insights on modelling organic aerosol. Finally, the statistics used for the model evaluation section is solid.

I have several minor suggestions, mainly on the terminology and the analysis used throughout the paper, and two major comments, before I can recommend final publication to ACP:

**Major comments:**
1) The large amount of CTM simulations are very helpful to explore the influence of the perturbed parameters on the modelled OA fraction, and I congratulate the authors for such analysis. However, I believe that it is important to mention (i.e. in the introduction, methodology or conclusion sections) that the perturbed space explored here is embedded in the parent VBS scheme that has been adopted. Therefore, the results should be interpreted in the light of the original VBS blueprint used by the author (e.g. 1D VBS, among other details). For instance: what will happen to the whole perturbed parameter space, if the original VBS will account for processes such as Fragmentation/Formation of HOMs/Autoxidation or for different kernel distributions of the oxidation products (e.g. different number of oxygen additions during the first oxidation steps, and therefore different allocation of the oxidation products across the volatility bins)?
I am not asking for more runs, but I believe this is an important point that needs to be considered.

2) The analysis of the diurnals is suggesting a strong influence of the meteorological parameters on the OA concentrations, likely of the boundary layer height diagnosed by the model. Even for the model

runs with the best overall statistics and optimized scaling factors, there is a positive bias during the early morning hours. This is an important result. Uncertainties in the diurnal emissions factors are sound, but I think the work will benefit from more analysis on the meteorological side. Are there any radiosondes measurements available to better understand/evaluate the model vertical output (see my additional comments below for the Results section)? In addition, how are the synoptic conditions of the pre-monsoon season reproduced by the model?

**Minor comments:**

**Abstract**

Abstract, line 28: Suggestion: I would replace "POA production and aging" with: "emissions of POA and its chemical evolution".

Abstract, line 29: Suggestion: "The main disadvantage is its complexity". I would not consider the complexity of the VBS a disadvantage per se, and I would suggest removing that sentence (I think it is already clear stated that the high numbers of parameters are difficult to probe).

Abstract, line 40: Suggestion: "in two periods: the full period (1st -29th May) and the period 14:00- 16:00 hrs local time, 1st-29th May". One period and one sub-period?

**Introduction**

Introduction, Line 50: Suggestion: I would move the sentence beginning at this line in the following paragraph. It is a bit detached from its current paragraph.

Introduction, Line 58: Please consider including Ghosh et al., 2021 who has recently applied the online version of the WRF-CHIMERE model over the same domain, i.e. the Indo-Gangetic Plain.

Introduction, Line 64: Please consider replacing "POA processes" with "OA processes".

Introduction, Line 70: Suggest using a more updated and theoretically oriented reference for the definition of the VBS domain, e.g. Bianchi et al., 2019.

Introduction, Line 73: Please consider rephrasing "The resulting low volatility oxidized organic vapors can condense to produce oxidized primary organic aerosols (oPOA) (Shrivastava et al., 2008)" with ". The resulting low volatility oxidized organic vapors can condense to produce secondary organic aerosol (SOA) (Shrivastava et al., 2008)". I think the model assigns those products to SOA (?). (oPOA is considered secondary, and there is no heterogeneous chemistry accounted for, or?).

Introduction, Line 92: "the partitioning of matter between gaseous and particulate phases, and the chemical aging of POA". Suggest rephrasing the sentence as: "The partitioning of organic matter and the evolution of POA".

**Methodology**

Methodology: How long was the spin-up period? In addition, was WRF nudged towards some reanalysis datasets (ERA5 or GFS)?

Methodology: What is the vertical resolution of the model and the first model layer height?

Methodology: Which scheme was used for the PBLH in the model? Please consider adding this information (see also my additional comment below).

Methodology, Line 121: "SOA components, each covering 4 volatility bins". At which C* (and T)? Please add such information. Do they cover the SVOC range, or also the LVOC/ELVOC range? In addition, it would be nice the report the IVOC yields with their oxidation pathways in the Methodology section.

Methodology, Line 130: "To investigate the impact of these assumptions on the model predictions, we have modified the model code so that the VBS emissions, the oxygenation rates and VBS reaction rates, can be directly controlled via namelist options". This is very nice.

Methodology, Line 134: "The volatility distribution of biomass burning". Suggest rephrasing with "The volatility distribution of open biomass burning" so to differentiate from the residential biomass burning sector contained in EDGAR-HTAP, which has different patterns and estimation methods (this is already specified in Table 1, and I suggest correcting it also in Table 2 accordingly).

Methodology, Line 137: "The volatility distribution for anthropogenic emissions is also multiplied by a scaling factor of 3". For anthropogenic emissions, I suggest adding also Denier van der Gon et al., 2015 which proposed a revisited emission inventory for the residential wood combustion sector based on a bottom-up approach. The revisited emission inventory, which try to account for the semi volatile components of the primary organic matter fraction, was a factor 2-3 higher compare to previously used emission inventories.

Methodology, Line 140 "Anthropogenic emissions are derived from the EDGAR-HTAP". Is there any spatial interpolation performed by the model on top of the original EDGAR-HTAP emission datasets? Please add. What was the projection used in the model?

Methodology, Line 163 ''PBLH data were sourced from ECMWF ERA5 with 0.25 deg". Why not directly from WRF at 15km? After further reading, I have realized that the aim is to compare WRF and ERA5. Are there any direct radiosondes measurements available?

Methodology, Figure 1: Is this an average over the whole period? Please add. I would actually consider adding the model topography instead of the $PM_1$ concentrations.

Methodology, Line 202: I am not sure I have understood this passage: The IVOCs are included by multiply by a factor of 1.5 the non-volatile OA mass (0.2+0.5+0.8). Several models have implemented such scaling factor (Koo et al., 2014; Tsimpidi et al., 2010) on top of the already increased emission inventory (i.e. by a factor of 3 to include SVOC based on partitioning theory calculation), so that the additional organic mass added to the model runs is 7.5 (i.e. 3 + 3*1.5). Is this the case also for this application? In addition, are the IVOCs factors, when increased or decreased during the sensitivity tests, applied to all the EDGAR sectors (i.e. one factor for all the sectors)? Please specify.

Methodology, Line 209: Are those factors, i.e. the parameter number 5 (i.e. 0.1 - 4), applied on top of the emissions inventory as described in the paragraph at line 137? I.e. a value of one would correspond to the original emissions inventory (i.e. increased by a factor of 3).

Methodology, Line 230, Table2. What are the units of the anthropogenic VBS ageing rate and of the open biomass burning VBS ageing rate? Please add.

**Results and discussions**

Results and discussions, Line 255: "the VBS setup will affect OA concentrations and PM, with no implications to inorganic aerosols or gaseous species". Would not the different oxidation schemes, and amount of organic precursors available such as SVOCs, IVOCs and VOC, alter the overall radical budget (e.g. $HO_x$ concentrations), and therefore also the inorganic and gas-phase chemistry?

Results and discussions, Line 267: "Total_OM". I guess it refers only to the modelled particle phase?

Results and discussions, Line 289: "we identified two potential structural errors in the WRF-Chem outputs, the early morning peak and the late evening low concentrations". This is an interesting part of the model evaluation section:

- I would suggest plotting also the diurnal of CO, which is probably a better candidate then OA for identifying the performance of the PBLH in the model. I think a diurnal of CO is already included in Figure S5 but I am not sure I have understood it properly (I would suggest improving the readability of Figure S5, mainly the choice of the colors. I think Figure S5 is also not referenced in the manuscript). It seems model data are reported every three-hours (?) and have a more flat profile compared to the observations (?). How this relate to the diurnal of OA? Along this line, why do the diurnals of O:C decrease in the model during daytime considering that daytime $O_3$ concentrations are over predicted ?

- For the analyzed period, it would be nice to know which one of the emissions sector in the EDGAR-HTAP dataset is the dominant one in that area, and what is the relative contribution of each of the sources.

- Is there any factor analysis on the AMS data to compare with? How are the modelled and the observational-based OA components (i.e. HOA and OOA) reproduced? Is it an overestimation related to the primary or to the secondary fraction? In addition, how will the model resolution (i.e. 15km) affect the transport and therefore the equilibration time between the particle and gas-phase organic material?

- I am not sure how much the comparison between ERA5-PBLH and WRF-PBLH could help here. Following one of my previous comment, I think it would be beneficial to clarify which parameterization has been used in WRF for the PBLH. ERA5 is based on the Richardson number method. How about those WRF runs? In addition, why the wind direction is not reported in Figure S4?

Model evaluation, line 340: "these model runs have low SVOC volatility distribution". Does this mean low factors (e.g. towards 0.05) and therefore a more volatile distribution (e.g. first panel in Figure S2)? Please add more details in the manuscript.

Model evaluation, Figure 5: This is a very nice plot, and clearly elucidates the various performance of the model runs. If possible, I would consider adding short acronyms for each of the 5 parameters at the pentagon's vertices (at least for one pentagon). It would highly facilitate the comprehension of the panel. In addition, even though provided on the github link, I would suggest to include a supplementary table were the 5 model's parameters are reported for the TRAIN 127, 121 126, 036, 117, 104, 115, 119 and 058. This will facilitate the use of such values by the modelling community. Finally, at least for TRAIN127 and TRAIN120, I would suggest producing in the manuscript a plot similar to Figure S2 where all the parameters are reported as insets on top of the volatility distribution plot associated to the final scaling factors. Again, this will highly facilitate the use of such parameters.

**Conclusions**

Conclusion, line 493: "low SVOC scaling". A more volatile distribution?

**References**

Bianchi, F., Kurtén, T., Riva, M., Mohr, C., Rissanen, M.P., Roldin, P., Berndt, T., Crounse, J.D., Wennberg, P.O., Mentel, T.F., Wildt, J., Junninen, H., Jokinen, T., Kulmala, M., Worsnop, D.R., Thornton, J.A., Donahue, N., Kjaergaard, H.G., Ehn, M., 2019. Highly Oxygenated Organic Molecules (HOM) from Gas-Phase Autoxidation Involving Peroxy Radicals: A Key Contributor to Atmospheric Aerosol. Chem. Rev. 119, 3472–3509. https://doi.org/10.1021/acs.chemrev.8b00395

Denier van der Gon, H.A.C., Bergström, R., Fountoukis, C., Johansson, C., Pandis, S.N., Simpson, D., Visschedijk, A.J.H., 2015. Particulate emissions from residential wood combustion in Europe – revised estimates and an evaluation. Atmos. Chem. Phys. 15, 6503–6519. https://doi.org/10.5194/acp-15-6503-2015

Ghosh, S., Verma, S., Kuttippurath, J., Menut, L., 2021. Wintertime direct radiative effects due to black carbon (BC) over the Indo-Gangetic Plain as modelled with new BC emission inventories in CHIMERE. Atmos. Chem. Phys. 21, 7671–7694. https://doi.org/10.5194/acp-21-7671-2021

Koo, B., Knipping, E., Yarwood, G., 2014. 1.5-Dimensional volatility basis set approach for modeling organic aerosol in CAMx and CMAQ. Atmospheric Environment 95, 158–164. https://doi.org/10.1016/j.atmosenv.2014.06.031

Tsimpidi, A.P., Karydis, V.A., Zavala, M., Lei, W., Molina, L., Ulbrich, I.M., Jimenez, J.L., Pandis, S.N., 2010. Evaluation of the volatility basis-set approach for the simulation of organic aerosol formation in the Mexico City metropolitan area. Atmos. Chem. Phys. 22.

---

## Author Comment (AC1)

Reply to reviewer's comments:

Reviewer 1 comments:

Reply to the reviewer in red.

1) The large amount of CTM simulations are very helpful to explore the influence of the perturbed parameters on the modelled OA fraction, and I congratulate the authors for such analysis. However, I believe that it is important to mention (i.e. in the introduction, methodology or conclusion sections) that the perturbed space explored here is embedded in the parent VBS scheme that has been adopted. Therefore, the results should be interpreted in the light of the original VBS blueprint used by the author (e.g. 1D VBS, among other details). For instance: what will happen to the whole perturbed parameter space, if the original VBS will account for processes such as Fragmentation/Formation of HOMs/Autoxidation or for different kernel distributions of the oxidation products (e.g. different number of oxygen additions during the first oxidation steps, and therefore different allocation of the oxidation products across the volatility bins)?

I am not asking for more runs, but I believe this is an important point that needs to be considered.

Indeed this is an important comment and it has been added in the methodology as follows: It is worth mentioning that the perturbed space explored here is embedded in the parent VBS scheme that has been adopted. There have been a large number of developments in, and variants of, the VBS aiming to address particular questions related to SOA formation at various levels of complexity (for example, the mechanistic measurement-constrained radical 2D-VBS examining the role of ELVOC and ULVOC in new particle formation; (Zhao et al., 2020; 2021). In the current study, our implementation has been developed from the VBS version available in the distribution version of WRF-Chem and our results should be interpreted in the context of the structural capabilities and limitations therein. More information about the VBS distributions and parameter space setup is in section S1 in the supplementary material.

Zhao, B., Shrivastava, M., Donahue, N. M., Gordon, H., Schervish, M., Shilling, J. E., Zaveri, R. A., Wang, J., Andreae, M. O., Zhao, C., Gaudet, B., Liu, Y., Fan, J., and Fast, J. D.: High concentration of ultrafine particles in the Amazon free troposphere produced by organic new particle formation, Proc Natl Acad Sci U S A, 117, 25344-25351, 10.1073/pnas.2006716117, 2020.
Zhao, B., Fast, J. D., Donahue, N. M., Shrivastava, M., Schervish, M., Shilling, J. E., Gordon, H., Wang, J., Gao, Y., Zaveri, R. A., Liu, Y., and Gaudet, B.: Impact of Urban Pollution on Organic-Mediated New-Particle Formation and Particle Number Concentration in the Amazon Rainforest, Environ Sci Technol, 55, 4357-4367, 10.1021/acs.est.0c07465, 2021.

2) The analysis of the diurnals is suggesting a strong influence of the meteorological parameters on the OA concentrations, likely of the boundary layer height diagnosed by the model. Even for the model runs with the best overall statistics and optimized scaling factors, there is a positive bias during the early morning hours. This is an important result. Uncertainties in the diurnal emissions factors are sound, but I think the work will benefit from more analysis on the meteorological side. Are there any radiosondes measurements available to better understand/evaluate the model vertical output (see my additional comments below for the Results section)? In addition, how are the synoptic conditions of the pre-monsoon season reproduced by the model?

We agree with the reviewer. There are no radiosondes measurements available to extend the analysis at the location and time period of our study. We have, in response to a question from reviewer #2, included comparisons of wind direction, speed, temperature and relative humidity between the model

and measurements for the study site. Hopefully these will help readers to evaluate the local model performance, even if we cannot provide information on the larger scale performance of the model.

**Minor comments:**

**Abstract**

Abstract, line 28: Suggestion: I would replace "POA production and aging" with: "emissions of POA and its chemical evolution". The authors replaced the phrase as suggested.

Abstract, line 29: Suggestion: "The main disadvantage is its complexity". I would not consider the complexity of the VBS a disadvantage per se, and I would suggest removing that sentence (I think it is already clear stated that the high numbers of parameters are difficult to probe). The line has been edited as follows: However, the evaluation of model uncertainty and the optimal model parameterisation maybe expensive to probe using only WRF-Chem simulations.

Abstract, line 40: Suggestion: "in two periods: the full period (1st -29th May) and the period 14:00-16:00 hrs local time, 1st-29th May". One period and one sub-period? Yes, that is correct. The line has been edited: The full period and one sub-period.

**Introduction**

Introduction, Line 50: Suggestion: I would move the sentence beginning at this line in the following paragraph. It is a bit detached from its current paragraph. The change has been done.

Introduction, Line 58: Please consider including Ghosh et al., 2021 who has recently applied the online version of the WRF-CHIMERE model over the same domain, i.e. the Indo-Gangetic Plain. The reference has been added

Introduction, Line 64: Please consider replacing "POA processes" with "OA processes". The change has been done.

Introduction, Line 70: Suggest using a more updated and theoretically oriented reference for the definition of the VBS domain, e.g. Bianchi et al., 2019. The reference has been added

Introduction, Line 73: Please consider rephrasing "The resulting low volatility oxidized organic vapors can condense to produce oxidized primary organic aerosols (oPOA) (Shrivastava et al., 2008)" with ". The resulting low volatility oxidized organic vapors can condense to produce secondary organic aerosol (SOA) (Shrivastava et al., 2008)". I think the model assigns those products to SOA (?). (oPOA is considered secondary, and there is no heterogeneous chemistry accounted for, or?). The change has been done.

Introduction, Line 92: "the partitioning of matter between gaseous and particulate phases, and the chemical aging of POA". Suggest rephrasing the sentence as: "The partitioning of organic matter and the evolution of POA". The change has been done.

**Methodology**

Methodology: How long was the spin-up period? In addition, was WRF nudged towards some reanalysis datasets (ERA5 or GFS)? A spin-up period of 11 days was used, from 19th April to 1st May. The meteorological driving fields were taken from ERA5 reanalysis data. Spectral nudging of the uv wind parameters, temperature and geopotential height variables to these, above model level 18 and for wavelengths greater than 950km, was used. We have added this information to the paper.

Methodology: What is the vertical resolution of the model and the first model layer height?

The model domain had 38, variable height and terrain following, model levels, up to a pressure of 50 hPa. The first model layer has a mean height of 59m over Delhi (and a mean height of 56m over the whole model domain). This information has been added to the paper.

Methodology: Which scheme was used for the PBLH in the model? Please consider adding this information (see also my additional comment below). The YSU PBL scheme was used, we added a note on this in the paper.

Methodology, Line 121: "SOA components, each covering 4 volatility bins". At which C\* (and T)? Please add such information. Do they cover the SVOC range, or also the LVOC/ELVOC range? In addition, it would be nice the report the IVOC yields with their oxidation pathways in the Methodology section. The 4 SOA volatility bins have C\* values (at 298 K) of 1, 10, 100, and 1000  $\mu$ g/m3, only covering the SVOC range. The yields are given in Table 2 of Tsimpidi et al (2010); we have added the details to the paper and we have also added a table containing the relevant yields to our supplementary material too. The IVOC compounds are included in the standard VBS scheme.

Methodology, Line 130: "To investigate the impact of these assumptions on the model predictions, we have modified the model code so that the VBS emissions, the oxygenation rates and VBS reaction rates, can be directly controlled via namelist options". This is very nice. Thank you, we appreciate your comment.

Methodology, Line 134: "The volatility distribution of biomass burning". Suggest rephrasing with "The volatility distribution of open biomass burning" so to differentiate from the residential biomass burning sector contained in EDGAR-HTAP, which has different patterns and estimation methods (this is already specified in Table 1, and I suggest correcting it also in Table 2 accordingly). The change has been done.

Methodology, Line 137: "The volatility distribution for anthropogenic emissions is also multiplied by a scaling factor of 3". For anthropogenic emissions, I suggest adding also Denier van der Gon et al., 2015 which proposed a revisited emission inventory for the residential wood combustion sector based on a bottom-up approach. The revisited emission inventory, which try to account for the semi volatile components of the primary organic matter fraction, was a factor 2-3 higher compare to previously used emission inventories. The reference has been added to the manuscript.

Methodology, Line 140 "Anthropogenic emissions are derived from the EDGAR-HTAP". Is there any spatial interpolation performed by the model on top of the original EDGAR-HTAP emission datasets? Please add. What was the projection used in the model? The EDGAR-HTAP emission dataset is interpolated onto the model grid, using the anthro\_emiss tool provided by NCAR. No further spatial processing of the EDGAR-HTAP emission dataset was carried out. Within Delhi the SAFAR-India emission dataset is used. The model domain projection is Lambert Conformal.

Methodology, Line 163 ''PBLH data were sourced from ECMWF ERA5 with 0.25 deg". Why not directly from WRF at 15km? After further reading, I have realized that the aim is to compare WRF and ERA5. Are there any direct radiosondes measurements available? Unfortunately we do not have direct radiosonde measurements of the boundary layer height in this location.

Methodology, Figure 1: Is this an average over the whole period? Please add. I would actually consider adding the model topography instead of the PM1 concentrations. Figure 1 has been updated with the topography data.

Methodology, Line 202: I am not sure I have understood this passage: The IVOCs are included by multiply by a factor of 1.5 the non-volatile OA mass (0.2+0.5+0.8). Several models have implemented such scaling factor (Koo et al., 2014; Tsimpidi et al., 2010) on top of the already increased emission inventory (i.e. by a factor of 3 to include SVOC based on partitioning theory calculation), so that the additional organic mass added to the model runs is 7.5 (i.e. 3 + 3\*1.5). Is this

the case also for this application? In addition, are the IVOCs factors, when increased or decreased during the sensitivity tests, applied to all the EDGAR sectors (i.e. one factor for all the sectors)? Please specify. The IVOCs are calculated from the final emission dataset, which includes all EDGAR sectors. In our study the base scaling factor for IVOCs (0.2+0.5+0.8=1.5) that gives us our initial emission amount, is scaled by another factor, between 0-3, to probe the sensitivity of the model to the abundance of IVOCs.

These details have been added to the paper.

Methodology, Line 209: Are those factors, i.e. the parameter number 5 (i.e. 0.1 - 4), applied on top of the emissions inventory as described in the paragraph at line 137? I.e. a value of one would correspond to the original emissions inventory (i.e. increased by a factor of 3). This is correct.

Methodology, Line 230, Table2. What are the units of the anthropogenic VBS ageing rate and of the open biomass burning VBS ageing rate? Please add. The VBS ageing rates are given in cm3 molec-1 s-1. We have added these units to Table 2.

**Results and discussions**

Results and discussions, Line 255: "the VBS setup will affect OA concentrations and PM, with no implications to inorganic aerosols or gaseous species". Would not the different oxidation schemes, and amount of organic precursors available such as SVOCs, IVOCs and VOC, alter the overall radical budget (e.g. HOx concentrations), and therefore also the inorganic and gas-phase chemistry? The gas-phase reactions which generate the SOA components, and age the VBS components, in this study do not consume OH, and so do not alter the overall radical budget. We chose to do this so that we could probe the model OA concentration and PM response only. It is worth mentioning that we did not want to introduce a coupling between the scheme and oxidants so as to introduce a second order feedback that we could not constrain.

We carried out some preliminary studies on the impact of including OH consumption (or not) on the atmospheric chemistry while setting up this study. We include below plots from these preliminary studies ('High' aging indicates a VBS reaction rate of  $1 \times 10^{-11}$  cm3 molec-1 s-1; 'low' aging indicates a VBS reaction rate of  $1 \times 10^{-12}$  cm3 molec-1 s-1), to give some guidance on how this impacts gas-phase chemistry. We found that including OH consumption could lead to 10-25% reduction in OH concentrations, depending on reaction rate. The impact on local ozone concentrations was not large (and not linear).

Results and discussions, Line 267: "Total\_OM". I guess it refers only to the modelled particle phase? Yes, this has been clarified in the manuscript.

Results and discussions, Line 289: "we identified two potential structural errors in the WRF-Chem outputs, the early morning peak and the late evening low concentrations". This is an interesting part of the model evaluation section:

- I would suggest plotting also the diurnal of CO, which is probably a better candidate then OA for identifying the performance of the PBLH in the model. I think a diurnal of CO is already included in Figure S5 but I am not sure I have understood it properly (I would suggest improving the readability of Figure S5, mainly the choice of the colors. I think Figure S5 is also not referenced in the manuscript). It seems model data are reported every three-hours (?) and have a more flat profile compared to the observations (?). How this relate to the diurnal of OA? Along this line, why do the diurnals of O:C decrease in the model during daytime considering that daytime O3 concentrations are over predicted ? Yes, the modelled CO concentrations are flatter than the CO observations. This is opposite to what is observed with OA, where, in general, OA observations are higher than the modelled OA. Figure S5 has now been referenced in the manuscript and a description has been added. We have also adjusted the colours in Figure S5 to make it more readable.

- For the analyzed period, it would be nice to know which one of the emissions sector in the EDGAR-HTAP dataset is the dominant one in that area, and what is the relative contribution of each of the sources.

- Is there any factor analysis on the AMS data to compare with? How are the modelled and the observational-based OA components (i.e. HOA and OOA) reproduced? Is it an overestimation related to the primary or to the secondary fraction? In addition, how will the model resolution (i.e. 15km) affect the transport and therefore the equilibration time between the particle and gas-phase organic material? Yes, there is a PMF analysis that we performed from the same dataset along with other datasets (Reyes-Villegas et al., 2021). In the following figure we highlight with a red square the PMF results that correspond to the data of the current study. We can see a morning peak around 9-10 hrs, which is higher in the secondary organic aerosol (more oxidised oxygenated OA – MO-OOA). This suggests that the large morning peak of the modelled organic mass that we observed in our diurnal analysis corresponds to the potential structural errors in the model, as mentioned in the manuscript.

---

## Author Response (AR2)

Simulating organic aerosol in Delhi with WRF-Chem using the VBS approach: Exploring model uncertainty with a Gaussian Process emulator

The authors appreciate the reviewer comments, the response is in red

The authors have successfully addressed most of the reviewer's questions. I do not have any major comments. However, there are a few minor comments, primarily concerning some clarifications and discussions that were provided in the response to reviewers but not included in the manuscript. Including these points in the manuscript would be useful to the community.

1) Line 95-97: The manuscript now clearly mentions that this study does not perturb SOA parameters. However, it is still missing a statement acknowledging that this study also assumes that dry and wet deposition simulation uncertainties are not important in Delhi (or assumes that the model accurately represents these processes).

We have edited the paragraph as follows: In this parameterisation we explore the perturbation to the chosen anthropogenic POA and biomass burning POA parameters that would be needed to give the best fit to the observed OA. We are not perturbing the SOA parameters from the base case nor the dry and wet deposition simulation uncertainties, analysis that is out of scope of this work. We also appreciate that there will be sensitivity to the deposition rate of OA components. We have focused our study on the sensitivity of the OA production processes at a constant deposition rate within WRF-Chem allowing reasonable conclusions about the plausible range of the other parameters to be drawn notwithstanding this limitation.

2) Line 145-149: While this information was provided in the response to reviewers, it would be beneficial for readers and the community to include the description of OM/OC ratio, non-oxygen mass, and oxygen mass treatments in emissions before applying the SVOC scaling factor, in the main text.

As suggested, we added the following text to the manuscript: Before applying the scaling factor we assumed a ratio of matter mass to carbon mass of 1.4, dividing the emission inventory matter mass by this to obtain the carbon mass. Within the model each VBS compound is stored as two variables, the oxygen part and the 'non-oxygen' part. When adding the emissions we multiply the carbon mass by 1.17 to get the 'non-oxygen' mass (carbon, plus other atoms), and by 0.08 to get the oxygen mass. These scaling factors were taken from Shrivastava et al. (2011). We then apply the SVOC scaling factor, and volatility distribution, to give the final SVOC emission profile. The IVOC scaling factor is applied to the same base emissions to get the IVOC emission profile.

3) Line 201-213: I found that the author's response to reviewer #2 concerning this scalar variable would be valuable to the community who will use the method presented in this study. Please consider adding more discussion on why creating this variable is essential for the emulator to function correctly for future studies.

The following paragraph, taken from the response, has been added to the manuscript:

The scalar variable represents a sensible range of possible emitted volatility distributions. A method was needed, by which we could represent the variation of possible volatility distributions within the process emulator. The direct approach would be to include a scaling factor for each volatility bin as separate parameters. However, this would have greatly increased the complexity and size of our parameter space, and these parameters would not be independent of each other, leading to a lot of wasted parameter space, waste in the use of our limited computer resources available for the PPE simulations and the assumptions for our variance-based sensitivity analysis becoming invalid.

Instead, we used a simple reaction model, where each step in a fraction of each volatility bin would be 'aged' and moved to the next volatility bin. This approach also allowed us to include some simple partitioning, with aging process stopped for any condensed matter; replicating the behaviour of the model these distributions will later be injected into. Given that we used a simple, fractional, aging process, it would not be appropriate for us to try to relate it to a physical variable. We have included Figure S2 instead, which gives example volatility distributions through the range of this scalar value used in our study.

4) 276-277: I think this line should be removed from the revised manuscript.

The line has been removed as suggested.

5) Line 296-297: It would be helpful for the authors to incorporate the explanation for this model trend in the manuscript since it has already been provided in their response to the reviewers.

We have added the requested explanation: We explored a range of emission multipliers (both IVOC and SVOC scaling). These upper limits, which have been of an appropriate magnitude for previous studies in other locations using different emission datasets (e.g. Shrivastava et al. (2011)), turned out to be too high for our emission dataset, and these are the model runs which produced the very high OM mass loadings (rather than these being predominately caused by high oxidation rates). When the OM mass loading is high, more of the higher volatility (and, here, less aged) compounds condense into the condensed-phase. The VBS scheme we have used has only gas-phase reactions, and so once in the condensed-phase these compounds do not age further. This process leads to the lower mean O:C ratios that are observed here.

6) Line 496-499: Please include the limitations of this study that arise from considering only semi-volatile POA processes. This study does not account for perturbations in SOA parameters (enthalpy of vaporization, SOA yield, photolytic loss) and deposition processes. I believe these processes are more important in Delhi than aqueous chemistry, as IEPOX-SOA is significant for biogenic-influenced regions (Shrivastava et al., 2022) rather than urban regions.

The paragraph has been edited as follows: It is worth mentioning that these structural errors might also be related to representation of other organic aerosol processes not represented by the VBS approach. As mentioned early in the introduction, this study only considers semi-volatile POA processes, without accounting for perturbations in SOA parameters and deposition processes. Recent studies, for example, have examined particle-phase and multiphase chemistry in aqueous aerosols and clouds (Shrivastava et al., 2022), and reactions of SOA precursors with other radicals like chlorine relevant to Indian conditions (Gunthe et al., 2021). Future studies could be focused on studying other parameters such as deposition processes and the perturbations in SOA parameters.